# OFFLINE REINFORCEMENT LEARNING WITH CLOSED-LOOP POLICY EVALUATION AND DIFFUSION WORLD-MODEL ADAPTATION

## ABSTRACT

Generative models, particularly diffusion models, have been utilized as world models in offline reinforcement learning (RL) to generate synthetic data, enhancing policy learning efficiency. Current approaches either train diffusion models once before policy learning begins or rely on online interactions for alignment. In this paper, we propose a novel offline RL algorithm, Adaptive Diffusion World Model for Policy Evaluation (ADEPT), which integrates closed-loop policy evaluation with world model adaptation. It employs an uncertainty-penalized diffusion model to iteratively interact with the target policy for evaluation. The uncertainty of the world model is estimated by comparing the output generated with different noises, which is then used to constrain out-of-distribution actions. During policy training, the diffusion model performs importance-sampled updates to progressively align with the evolving policy. We analyze the performance of the proposed method and provide an upper bound on the return gap between our method and the real environment under the target policy. The results shed light on various key factors affecting learning performance. Evaluations on the D4RL benchmark demonstrate significant improvement over state-of-the-art baselines, especially when only suboptimal demonstrations are available – thus requiring improved alignment between the world model and offline policy evaluation.

## 1 INTRODUCTION

Offline Reinforcement Learning (RL) methods have garnered much attention recently (Levine et al., 2020; Prudencio et al., 2023), due to their abilities to train policies based on offline datasets collected by a behavior policy, rather than relying on costly and sometimes risky online interactions (Kiran et al., 2021). With only limited data of transitions or trajectories available, it is prone to overestimating the reward from out-of-distribution states and actions as the learned policy gradually deviates from the behavior policy used for collecting the data. This issue, known as the distributional shift (Kumar et al., 2019), is one of the key challenges in offline RL.

To address this, various solutions have been proposed, such as augmenting the dataset by building world models (Yu et al., 2021; Rigter et al., 2022; Matsushima et al., 2020), and policy regularization approaches (Kumar et al., 2019; Rashidinejad et al., 2021). The key idea of building world models is to learn the transition dynamics of the underlying Markov Decision Process (MDP). Once trained on the offline dataset, the policy can interact with the world model and generate additional synthetic trajectories. Existing works include world models generated by VAE (Ha & Schmidhuber, 2018; Hafner et al., 2023; Ozair et al., 2021), GAN (Eysenbach et al., 2022), Transformers (Janner et al., 2021), and more recently Diffusion (Ding et al., 2024a; Lu et al., 2023). However, most of these works either train a world model one-time prior to policy learning (Yu et al., 2020; Kidambi et al., 2020; Janner et al., 2019) or require additional online interaction data to adapt the world model (Kaiser et al., 2019; Hafner et al., 2019). Neither approach effectively mitigates distributional shifts, relying solely on offline data. Furthermore, the return gap between world models (e.g., diffusion models) and the real environment in such offline RL algorithms remains to be analyzed.

This paper proposes a novel approach to offline RL, integrating policy evaluation and world-model adaptation, namely the Adaptive Diffusion World Model for Policy Evaluation (ADEPT). ADEPT

encompasses two collaborative components: (i) a guided diffusion world model with uncertainty estimation, which generates uncertainty-penalized synthetic trajectories by evaluating the target policy step-by-step; and (ii) importance-sampled updates, based on the offline dataset, to align the world model with the evolving policy. These two components operate in a closed loop throughout training. Existing works in RL typically treat diffusion world models as synthesizers for data augmentation (Lu et al., 2023), policy representation (Wang et al., 2022; Chen et al., 2022), or multistep planners (Janner et al., 2022; Ding et al., 2024a). These approaches generate additional synthetic trajectories with a fixed world model throughout the policy improvement iterations. In contrast, ADEPT continually adapts the diffusion model using importance sampling with respect to the distributional shift between the target policy $\pi_k$ and the behavior policy $\pi_b$. The updated diffusion model is then used to evaluate $\pi_k$ for policy improvement, with a sequence of actions drawn from $\pi_k$ used as the conditional input to the world model. We note that ADEPT requires alterations between next-state generation using the guided diffusion world model and next-action sampling from the target policy.

Furthermore, we analyze the performance of the proposed ADEPT algorithm theoretically and empirically. We provide the bound of the return gap between the policy under actual environment and that under the diffusion world model, and show that the monotonic improvement of return can be guaranteed when the one-step policy update under the world model is larger than this bound. We further decompose the bound into three factors: model transition error, reward prediction error and policy shift, and discuss how ADEPT lowers these factors and addresses other significant issues such as distributional shift problem to enhance the performance. To the best of our knowledge, this is the first analysis for offline RL with diffusion world models. The proposed algorithm is evaluated on D4RL benchmark (Fu et al., 2020) in multiple MuJoCo environments. We notice that ADEPT works best with datasets consisting of mainly suboptimal demonstrations, where the distributional shift becomes more severe as target policy moves toward optimum. The results show that ADEPT significantly improves the baseline SAC method (Haarnoja et al., 2018), and largely outperforms other SOTA offline RL methods including model-free, model-based, and diffusion-based algorithms with up to 20% improvement on average.

The contributions of this work can be summarized as follows:

1. We propose a novel model-based offline RL algorithm, ADEPT, with closed-loop operations of policy evaluation and improvement based on an uncertainty-penalized diffusion model and world model adaptation with importance sampling.

2. We provide theoretical proof of bounding the return gap between the diffusion world model and actual environment under the same policy, and discuss how ADEPT narrows the bound and solves the key issues to enhance the performance.

3. We evaluate our method on the D4RL benchmarks and demonstrate significant improvement over the state-of-the-art baselines, especially on suboptimal datasets.

## 2 RELATED WORKS

**Offline RL.**   Offline RL faces the distributional shift problem (Kumar et al., 2020) due to data collected using a specific behavior policy. Various methods have been proposed to regularize an offline RL policy and address this issue. In particular, model-based offline RL methods such as MOReL (Kidambi et al., 2020), MOPO (Yu et al., 2020), RAMBO (Rigter et al., 2022) and COMBO (Yu et al., 2021) train an extra MLP to mimic the dynamics and develop different ways to penalize the reward or value function in unseen state and action pairs to address the out-of-distribution issues. Other model-free methods, including BCQ (Fujimoto et al., 2019), IQL (Kostrikov et al., 2021), CQL (Kumar et al., 2020), and TD3+BC (Fujimoto & Gu, 2021), add different regularization mechanisms that are defined on action or value function, forcing the policy to act more conservatively. Our proposed ADEPT framework can be combined with any of these offline RL algorithms.

**World Models for RL.**   The use of world models to generate synthetic data for RL was first proposed by Ha & Schmidhuber (2018), as an extension of traditional model-based RL methods, utilizing VAE and RNN for predicting environmental transitions. Following this approach, various world models with advanced capabilities of fitting desired distributions have been proposed, including convolutional U-networks (Kaiser et al., 2019), vector-quantized autoencoders (Ozair et al., 2021),

generative adversarial networks (Eysenbach et al., 2022), energy-based models (Boney et al., 2020), transformers (Janner et al., 2021), and diffusion (Ding et al., 2024a; Lu et al., 2023). These world models are mainly used for trajectory synthesis with limited adaptability.

**The Use of Diffusion Models in RL.** Diffusion is a state-of-the-art technique for generating synthetic samples of images, videos or texts (Ho et al., 2020). It was first introduced by Janner et al. (2022) as a multistep planner in offline RL, where the diffusion model directly generates trajectories that are used for decision-making. This is further extended to conditional actions (Ajay et al., 2022), meta-RL (Ni et al., 2023), hierarchical tasks (Li et al., 2023), multitask problems (He et al., 2023), multi-agent tasks (Zhu et al., 2023) and safe planning (Xiao et al., 2023). Diffusion models are also employed for policy expression (Wang et al., 2022; Chen et al., 2022), imitation learning (Hegde et al., 2024) and reward modeling (Nuti et al., 2023). Lu et al. (2023) adopted the diffusion model as a data synthesizer to generate additional synthetic data based on offline datasets before policy training. Later, a conditional diffusion world model was proposed to generate trajectories from current state and action, to support offline value-based RL (Ding et al., 2024a) and policy-based RL (Jackson et al., 2024). However, these methods don't fully solve the distributional shift issue since the shift problem in world model is not considered. Different from these existing works, our method ADEPT utilizes the essence of diffusion model to estimate uncertainty for policy regularization, and adapt the diffusion model through importance sampling inspired by previous methods in online RL(Wang et al., 2023). The theoretical analysis of the return gap between ADEPT and the actual environment under the optimal policy is also provided.

## 3 PRELIMINARIES

**Offline RL using World Models** We consider an unknown MDP, referred to as the environment. Supposing the MDP is fully observable with discrete time, it could be defined by the tuple $\mathcal{M} = (S, A, P, R, \mu_0, \gamma)$. $S$ and $A$ are the state and action spaces, respectively. $P(s'|s, a)$ is the transition probability with $s, s' \in S, a \in A$, and $R : S \times A \to \mathbb{R}$ is the reward function. $\mu_0$ is the initial state distribution and $\gamma$ is the discount factor. We consider an agent that acts within the environment based on a policy $\pi(a|s)$ repeatedly. In each time step $t$, the agent receives a state $s_t$ and samples an action via its policy $a_t \sim \pi(\cdot|s_t)$. The environment transforms into a new state $s_{t+1} \sim P(\cdot|s_t, a_t)$ and returns a reward $r_t = R(s_t, a_t)$. After a whole episode of interactions, a trajectory $\tau = (s_0, a_0, r_0, \ldots, s_{T-1}, a_{T-1}, r_{T-1}, s_T)$ will be generated, which contains states, actions and rewards of maximum length $T$. Based on that, the goal of RL is to learn an optimal policy $\pi^*$ to maximize the expectation of discounted cumulative rewards from this MDP: $\pi^* = \arg\max_\pi \mathbb{E}_{(s,a)\sim\pi}(\sum_{t=0}^{T-1} \gamma^t r_t)$.

Specifically in offline model-based RL, only a dataset of trajectories $\mathcal{D}$ is available, and a prediction model of the environment is introduced, denoted as world model $\hat{\mathcal{M}} = (S, A, \hat{P}, \hat{R}, \mu_0, \gamma)$, to improve sample efficiency for further learning and planning. Commonly, the world model learns a single-step transition approximating the real dynamics $P$ of the environment in a supervised method based on $\mathcal{D}$. Hence, once a world model has been trained, it could replace the real environment to generate synthetic trajectories. Similar to standard RL, an initial state $s_0$ is sampled first from datasets, and based on that the interactions start. After certain length of steps $H$, referred as horizon, a synthetic trajectory $\hat{\tau} = (\hat{s}_0, \hat{a}_0, \hat{r}_0, ..., \hat{s}_{H-1}, \hat{a}_{H-1}, \hat{r}_{H-1}, \hat{s}_H)$ is generated, in which $\hat{s}_{t+1} \sim \hat{P}(\cdot|\hat{s}_t, \hat{a}_t)$ and $\hat{r}_t = \hat{R}(s_t, a_t)$. These imaginary trajectories are added into the replay buffer for policy optimization.

**Diffusion Model** The purpose of the diffusion model is to learn an underlying data distribution $q(\boldsymbol{x}_0)$ from a dataset $\mathcal{D} = \{\boldsymbol{x}_i\}$. In DDPM (Nichol & Dhariwal, 2021), the synthetic data generation is conducted by denoising real data $\boldsymbol{x}_0$ from noises $\mathcal{N}(\boldsymbol{0}, \boldsymbol{I})$ with $K$ steps. During training, a forward process $q(\boldsymbol{x}_k|\boldsymbol{x}_{k-1}) = \mathcal{N}(\sqrt{\alpha_{k-1}}\boldsymbol{x}_{k-1}, \sqrt{1-\alpha_{k-1}}\boldsymbol{I})$ is adopted to add noise on real data step by step, leading the final distribution towards Gaussian noises. The diffusion model learns a parameterized reverse process $p_\theta(\boldsymbol{x}_{k-1}|\boldsymbol{x}_k) = \mathcal{N}(\mu_\theta(\boldsymbol{x}_k), \Sigma_k)$ to denoise the real data from the Gaussian noise $\mathcal{N}(\boldsymbol{0}, \boldsymbol{I})$. By defining $\bar{\alpha}_k = \prod_{i=1}^k \alpha_i$, $\mu_\theta$ and $\Sigma_k$ can be rewritten as follows:

$$\mu_\theta(\boldsymbol{x}_k) = \frac{1}{\sqrt{a_k}}\left(\boldsymbol{x}_k - \frac{\beta_k}{\sqrt{1-\bar{\alpha}_k}}\boldsymbol{\epsilon}_\theta(\boldsymbol{x}_k, k)\right) \text{ and } \Sigma_k = \beta_k \frac{1-\bar{\alpha}_{k-1}}{1-\bar{\alpha}_k}\boldsymbol{I}. \tag{1}$$

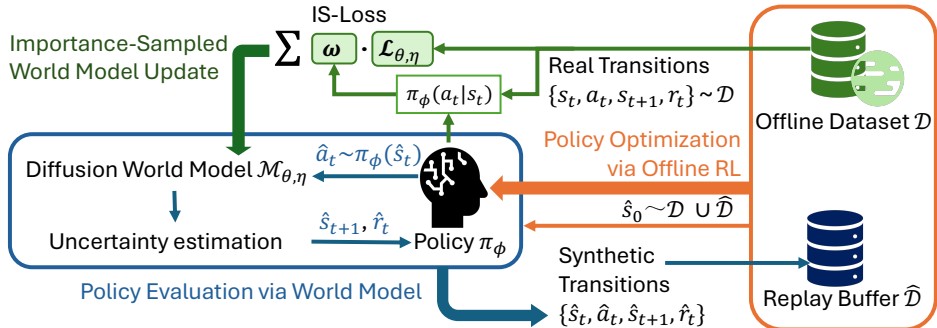

Figure 1: An overview of our ADEPT algorithm. Thin arrows denote the data flows, while thick arrows represent the key steps in ADEPT. It iteratively leverages an uncertainty-penalized diffusion world model to directly evaluate the target policy with actions drawn from it and optimize the policy with the collected data, and then performs an importance-sampled world model update to adaptively align the world model with the updated policy.

Here, $\epsilon_\theta$ is the parameterized noise prediction model to be trained. Therefore, the loss function of the diffusion model is defined as: $L(\theta) = \mathbb{E}_{k \sim [1,K], x_0 \sim q, \epsilon \sim N(\mathbf{0}, \mathbf{I})}(\|\epsilon - \epsilon_\theta(\boldsymbol{x}_k, k)\|^2)$, where $\epsilon$ is the real noise added in each step. When generating the synthetic data $\hat{x}_0$, beginning with $\hat{x}_K$ sampled from $\mathcal{N}(\mathbf{0}, \mathbf{I})$, $\epsilon_\theta$ predicts noise possibly added in each step until reaching $\hat{x}_0$. Specifically in this work, the noise prediction model is conditioned by the current state and action to predict the next state, denoted as $\epsilon_\theta(s_{t+1,k}, (s_t, a_t), k)$. We adopt the classifier-free method (Ho & Salimans, 2022) to train the diffusion model.

## 4 METHODOLOGY

In this section, we illustrate the details of ADEPT. As shown in Figure 1, the two key components in ADEPT are policy evaluation on a guided diffusion world model and importance-sampled world model updates. They work in a closed-loop operation through the whole training process, The explanations of these two components are covered in Section 4.1 and Section 4.2, respectively. In Section 4.3, we provide theoretical derivation to bound the return gap of the proposed method.

### 4.1 POLICY EVALUATION USING GUIDED DIFFUSION MODEL

Before the policy iteration, we first utilize the offline dataset to initialize the guided diffusion world model $\hat{\mathcal{M}}_{\theta,\eta}$, consisting of noise prediction model $\epsilon_\theta$ and a two-layer MLP $r_\eta$, to simulate the conditional distribution of the one-step transition $P(s_{t+1}|s_t, a_t)$ and reward function $R(s_t, a_t)$, respectively. We denote the simulated transition probability via $\epsilon_\theta$ as $P_\theta$. The offline dataset is normalized by linearly mapping each dimension of the state and action spaces to have 0 mean and 1 standard deviation. While training the diffusion model, we sample a minibatch of tuples $(s_t, a_t, r_t, s_{t+1})$ consisting of the state, action, reward, and next state from the normalized offline datasets in each iteration. As mentioned in Section 3, we follow DDPM and adopt the classifier-free method to learn the conditional probabilities of the state transition based on current state $s_t$ and action $a_t$. Since the reward function plays a significant role in RL training, the model should emphasize more on its accuracy and decouple the reward function from state transition dynamics. Therefore, we separately train $r_\eta(s_t, a_t, s_{t+1})$ to predict the reward function and the terminal signal. Introducing $s_{t+1}$ as an extra input significantly improves the accuracy of reward prediction, and in evaluation $s_{t+1}$ is replaced by $\hat{s}_{t+1}$, which is sampled from $P_\theta(\hat{s}_{t+1}|s_t, a_t)$.

However, a naive model-based RL method directly using initialized $\hat{\mathcal{M}}_{\theta,\eta}$ for planning could suffer from the inaccuracy brought by out-of-distribution states and actions, *i.e.*, the model may overestimate the rewards in state-action pairs not included in the dataset. To deal with the distributional shift problem, previous methods mainly use the ensemble of learned models to estimate the uncertainty (Lowrey et al., 2019; Yu et al., 2020; Kidambi et al., 2020). For the diffusion model, since

(a) Traditional World Model      (b) Adaptable World Model with Importance Sampling

Figure 2: Comparison of traditional world model and adaptable world model in ADEPT. Each curve refers to the trajectory expectation under a certain policy. Blue points refer to samples in the offline dataset, while darker points are given higher weight in loss calculation. Each circle indicates a distribution of the generated samples under the world model $\hat{\mathcal{M}}$ with relatively lower uncertainties. The existing algorithms as shown in (a) use a fixed world model in offline RL. Our proposed approach, as illustrated in (b), works in a loop between policy evaluation and model adaption.

each output is denoised from Gaussian noise, the uncertainty could be directly estimated from the discrepancy of multiple denoised samples with the same condition:

$$d_\theta(s_t, a_t) = \max_{i,j} \|\hat{s}_{t+1,i} - \hat{s}_{t+1,j}\|, \tag{2}$$

in which $\hat{s}_{t+1,i}, \hat{s}_{t+1,j} \sim P_\theta(s_{t+1}|s_t, a_t)$. This uncertainty will be used to determine the terminal signal $d_t$, and be added as a penalty to the predicted reward $\hat{r}_t$.

---

**Algorithm 1:** Policy Evaluation using Guided Diffusion Model ($PE$)

---

Randomly select a state $s_t$ from $(s_t, a_t, r_t, s_{t+1}) \in \mathcal{D} \cup \hat{\mathcal{D}}$ as $\hat{s}_0$.
**for** $t = 0 \to H$ **do**
    $\hat{a}_t \sim \pi_\phi(\cdot|\hat{s}_t)$
    **for** $i = 0 \to N_d$ **do**
        $x_K \sim \mathcal{N}(\mathbf{0}, \mathbf{I})$
        **for** $k = K - 1 \to 0$ **do**
            $x_k \sim \mathcal{N}(\mu_\theta(x_{k+1}|\hat{s}_t, \hat{a}_t), \mathbf{\Sigma_k})$, in which $\mu_\theta$ and $\mathbf{\Sigma_k}$ are defined in equation 1.
        $\hat{s}_{t+1,i} = x_0$
    $d_\theta(\hat{s}_t, \hat{a}_t) = \max_{i,j} \|\hat{s}_{t+1,i} - \hat{s}_{t+1,j}\|$; $\hat{s}_{t+1} = average(\{\hat{s}_{t+1,i}\}_{i=0}^{N_d-1})$
    $\hat{r}_t = r_\eta(\hat{s}_t, \hat{a}_t, \hat{s}_{t+1}) - \lambda_r d_\theta(\hat{s}_t, \hat{a}_t)$; $d_t = \delta(d_\theta(\hat{s}_t, \hat{a}_t) > \lambda_0)$
    **If** $d_t == 1$ **then** break
Return the trajectory $\hat{\tau} = (\hat{s}_0, \hat{a}_0, \hat{r}_0, \hat{s}_1, \ldots \hat{s}_t, \hat{a}_r, \hat{r}_t, \hat{s}_{t+1})$

---

The process of policy evaluation and improvement based on the uncertainty-penalized diffusion model is summarized in Algorithm 1. After initialization, $\hat{\mathcal{M}}_{\theta,\eta}$ interacts with the current policy $\pi_\phi$ and generates data for evaluation, and saves it in the replay buffer $\hat{\mathcal{D}}$. At the beginning of each evaluating iteration, $\hat{\mathcal{M}}_{\theta,\eta}$ randomly samples a state from $\mathcal{D} \cup \hat{\mathcal{D}}$ to be the start state $\hat{s}_0$, even though it may appear as a middle state in the real trajectory. Based on the current state $\hat{s}_t$, an action $\hat{a}_t \sim \pi_\phi(\hat{s}_t)$ is sampled given target policy. Conditioned by $\hat{s}_t$ and $\hat{a}_t$, the diffusion model generates $N_d$ possible next states $\{\hat{s}_{t+1,i}\}_{i=0}^{N_d-1}$ after $K$ denoising steps each via $\epsilon_\theta$. Next, the average and discrepancy of the states are calculated as the next predicted state $\hat{s}_{t+1}$ and the estimated uncertainty, respectively. The final reward is a combination of $r_\eta(\hat{s}_t, \hat{a}_t, \hat{s}_{t+1})$ and uncertainty penalty. If the uncertainty is larger than a threshold $\lambda_r$ or the next state satisfies known termination conditions, a terminal signal $d_t$ will be activated. Such an iteration continues till when $d_t$ is true or $t$ reaches horizon $H$.

### 4.2 Importance-Sampled World Model Update

Once the policy is updated, there is a policy shift between $\pi_\phi$ and the behavior policy $\pi_\mathcal{D}$ that collected $\mathcal{D}$. The estimation from $\hat{\mathcal{M}}_{\theta,\eta}$ could lose accuracy under the new distribution brought by the policy update. To handle this, we adopt the importance-sampling technique to update $\hat{\mathcal{M}}_{\theta,\eta}$ with

offline dataset, guiding $\hat{\mathcal{D}}$ towards the accurate distribution under the current policy. This is achieved by re-weighting the loss function of multiple samples to reduce the discrepancy between $\pi_\phi$ and $\pi_\mathcal{D}$. Even if $\pi_\mathcal{D}$ is not available, it's not hard to estimate the behavior policy from the offline dataset via behavior cloning (BC) (Nair et al., 2018). For each transition $\{(s_{t_i}^i, a_{t_i}^i, r_{t_i}^i, s_{t_{i+1}}^i)\}_{i=1}^N$ in the training batch, given the importance weight as $\omega_i$, the individual loss function as $l_i(\theta, \eta)$, which is later defined in Algorithm 2, then the total loss $\mathcal{L}(\theta, \eta)$ of the whole training batch is calculated as:

$$\mathcal{L}(\theta, \eta) = \frac{1}{N} \sum_{i=1}^N \omega_i l_i(\theta, \eta) = \frac{1}{N} \sum_{i=1}^N \frac{\pi_\phi(a_{t_i}^i | s_{t_i}^i)}{\pi_\mathcal{D}(a_{t_i}^i | s_{t_i}^i)} l_i(\theta, \eta). \tag{3}$$

Generally, the state-action pairs that have more probabilities under the current policy are associated with a larger weight in loss calculation.

---

**Algorithm 2:** Framework for ADEPT algorithm

---

**Input:** offline dataset $\mathcal{D}$, diffusion world model $\hat{\mathcal{M}}_{\theta, \eta}$

Initialize target policy $\pi_\phi$, replay buffer $\hat{\mathcal{D}} = \varnothing$; Normalized the dataset $\mathcal{D}$

Initialize $\hat{\mathcal{M}}_{\theta, \eta}$ with $\mathcal{D}$ and $\pi_\mathcal{D}$ till convergence: $IWU(\pi_\mathcal{D}, \mathcal{D}, \theta, \eta)$

**while** *not converged* **do**

    **for** $j = 0 \to N_e$ **do**

        $\hat{\mathcal{D}} \leftarrow \hat{\mathcal{D}} \cup PE(\hat{\mathcal{M}}_{\theta, \eta}, \pi_\phi)$

    Sample $\mathcal{B} = \{(s_{t_i}^i, a_{t_i}^i, r_{t_i}^i, s_{t_{i+1}}^i)\}_{i=1}^{B_p} \sim \mathcal{D} \cup \hat{\mathcal{D}}$

    Update $\phi$ with $\mathcal{B}$ via offline RL methods

    $IWU(\pi_\phi, \mathcal{D}, \theta, \eta)$

---

**Subroutine:** Importance-Sampled World-Model Update ($IWU$):

Sample batch $\{(s_{t_i}^i, a_{t_i}^i, r_{t_i}^i, s_{t_{i+1}}^i)\}_{i=1}^{B_m} \sim \mathcal{D}$

**for** $i = 0 \to B_m$ **do**

    $k \sim Uniform(\{1, 2, \ldots, K\})$; $\epsilon \sim \mathcal{N}(\mathbf{0}, \boldsymbol{I})$

    $s_{noise} = \sqrt{\bar{\alpha}_k} s_{t_{i+1}}^i + \sqrt{1 - \bar{\alpha}_k} \epsilon$

    Get Importance-sampling weight $\omega_i$ under $\pi$ via equation 3

    $l_i = \|\epsilon - \epsilon_\theta(s_{noise}, (s_{t_i}^i, a_{t_i}^i), k)\|^2 + \|r_t - r_\eta(s_{t_i}^i, a_{t_i}^i, s_{t_{i+1}}^i)\|^2$

Calculate $\mathcal{L}(\theta, \eta)$ via equation 3 and take gradient step on it.

---

The complete training procedure of ADEPT is illustrated in Algorithm 2, in which policy evaluation and world-model update alternate iteratively until convergence. The meaning and selection of each hyperparameter are further discussed in the appendix. In this work, we adopt the traditional off-policy algorithm SAC (Haarnoja et al., 2018) as the offline RL method to show the performance with no extra policy regularization methods. The results of combining ADEPT with other model-free offline RL methods are included in the appendix.

## 4.3 RETURN GAP ANALYSIS

In this section, we give a sketch of our theory and display the detailed proof in the Appendix, which is inspired by previous works (Yu et al., 2020; Kidambi et al., 2020; Janner et al., 2019), especially for Lemma A.1 and Lemma A.2. However, the uniqueness of our analysis is that we not only quantify the effects of uncertainty-based penalty and truncation in our theory, but also justify the use of importance sampling for optimizing the world model on a tighter bound compared to previous methods. Finally, we summarize three sources of errors that determine the return gap, and demonstrate how ADEPT is different on dealing with them.

To show that the improvement on expected return $J(\pi) = \mathbb{E}_{(s_t, a_t) \sim \pi | \mathcal{M}} \sum_{t=0}^T \gamma^t r_t$ could be guaranteed when adopting diffusion model as the world model to optimize the policy $\pi$, we wish to provide a bound $C$ of the return gap between the performance of $\pi$ under the real environment and that under the diffusion world model. Unlike traditional model-based RL methods, ADEPT samples the initial state from the whole dataset and utilizes an uncertainty threshold to truncate the rollout. Thus, we first define $\Gamma(\pi)$ as the expected time step that the policy choose a state and action pair with a discrepancy

larger than $\lambda_0$:

$$\Gamma(\pi) = \mathbb{E}_{(s_t,a_t)\sim\pi|\mathcal{M}}[\min t|d_\theta(s_t, a_t) \le \lambda_0] \tag{4}$$

Next, instead of the whole trajectory, we focus on the return gap of the truncated parts. More specifically, we seek a bound C of the return gap between the real environment and the world model under the same policy:

$$J(\pi) - \gamma^{\Gamma(\hat{\pi})} J_{\Gamma(\hat{\pi})}(\pi) \ge \hat{J}_{\theta,\eta}(\pi) - C. \tag{5}$$

Here, $J(\pi)$ and $\hat{J}_{\theta,\eta}(\pi)$ are the expected returns of $\pi$ under the real environment and the uncertainty-penalized diffusion world model $\hat{\mathcal{M}}_{\theta,\eta}$, respectively. $J_{\Gamma(\pi)}(\pi) = \mathbb{E}_{(s_t,a_t)\sim\pi|\mathcal{M}}[\sum_{t=\Gamma(\pi)}^T \gamma^{t-\Gamma(\pi)} r_t]$ is the expectation of returns starting from the distribution of time step $\Gamma(\pi)$.

Once $C$ is obtained, we could guarantee policy improvement under the actual environment if the returns of $\pi$ under the world model are promoted by at least $C$. Furthermore, $C$ is expected to be expressed in terms of the error and uncertainty quantities of the world model. We denote the reward prediction error as $\hat{\varepsilon}_r$, the model transition error as $\hat{\varepsilon}_m$, and policy distributional shift error as $\hat{\varepsilon}_p$. Their detailed definitions are presented as follows:

**Definition 4.1.** We define $\hat{\varepsilon}_r(\pi)$ to be the maximal expectation of half the absolute difference between predicted reward and true reward under the policy $\pi$.

$$\hat{\varepsilon}_r(\pi) = \max_t \mathbb{E}_{(s_t,a_t)\sim\pi|\mathcal{M}}\left[\frac{1}{2}|R(s_t, a_t) - r_\eta(s_t, a_t)|\right]. \tag{6}$$

Although in practice the reward prediction MLP takes $\hat{s}_{t+1}$ as an extra input to improve accuracy, we can still consider the combination of diffusion model and MLP as the theoretical reward model: $r_{\eta,\theta}(s_t, a_t) = \sum_{s'} r_\eta(s_t, a_t|s') P_\theta(s'|s_t, a_t)$, which doesn't affect the derivation.

**Definition 4.2.** $\hat{\varepsilon}_m(\pi)$ is defined as the maximal expected total-variation distance (TV-distance) of the probabilities between predicted next state and true value under $\pi$.

$$\hat{\varepsilon}_m(\pi) = \max_t \mathbb{E}_{(s_t,a_t)\sim\pi|\mathcal{M}}\left[D_{TV}(P(s_{t+1}|s_t, a_t)\|P_\theta(s_{t+1}|s_t, a_t))\right], \tag{7}$$

**Definition 4.3.** $\hat{\varepsilon}_p(\pi)$ is defined as the maximal TV-distance of $\pi$ and the behavior policy $\pi_\mathcal{D}$ that collects the offline dataset. This error measures how the target policy $\pi$ has shifted from the behavior policy $\pi_\mathcal{D}$.

$$\hat{\varepsilon}_p(\pi) = \max_s D_{TV}(\pi(a|s)\|\pi_\mathcal{D}(a|s)). \tag{8}$$

Besides, we assume that the discrepancy $d_\theta(s_t, a_t)$ could be used as an uncertainty estimator.

**Assumption 4.4.** We assume $d_\theta(s_t, a_t)$ to be an admissible error estimator for both the model transition error and the reward prediction error, under appropriately selected parameters $\alpha_m$ and $\alpha_r$, respectively. It follows that for any policy $\pi$ during training, these conditions are satisfied:

$$\mathbb{E}_{(s_t,a_t)\sim\pi|\mathcal{M}}\left[D_{TV}(P(s_{t+1}|s_t, a_t)\|P_\theta(s_{t+1}|s_t, a_t))\right] \le \alpha_m \mathbb{E}_{(s_t,a_t)\sim\pi|\mathcal{M}}\left[d_\theta(s_t, a_t)\right], \tag{9}$$

$$\mathbb{E}_{(s_t,a_t)\sim\pi|\mathcal{M}}\left[\frac{1}{2}|R(s_t, a_t) - r_\eta(s_t, a_t)|\right] \le \alpha_r \mathbb{E}_{(s_t,a_t)\sim\pi|\mathcal{M}}\left[d_\theta(s_t, a_t)\right]. \tag{10}$$

While lacking theoretical guarantees, using the standard deviation to estimate the uncertainty have already been applied in many existing works(Rajeswaran et al., 2022; Kurutach et al., 2018; Yu et al., 2020), and supported by the experiments in the appendix. This assumption is mainly used as the justification of uncertainty penalty, and doesn't involve much in the derivation of return gap. With the errors and the assumption listed above, we provide the main theorem with a bound $C$ based on $\hat{\varepsilon}_m(\pi)$ and $\hat{\varepsilon}_r(\pi)$ to show the return gap under the true environment and the world model. Besides, we also provide a corollary with a softer bound $C'$ based on $\hat{\varepsilon}_m(\pi_\mathcal{D})$ and $\hat{\varepsilon}_r(\pi_\mathcal{D})$, and $\hat{\varepsilon}_p(\pi)$, which is mainly used by the previous algorithms.

**Theorem 4.5.** *Given $\hat{\varepsilon}_r$, $\hat{\varepsilon}_m$, the bound $C$ between the true return and the ADEPT model return can be expressed as follows:*

$$C = \sum_{t=0}^{\Gamma(\pi)-1} \gamma^t \left((2 - \frac{\lambda_r}{\alpha_r})\hat{\varepsilon}_r(\pi) + 2r_{max}(t+1)\hat{\varepsilon}_m(\pi)\right) \tag{11}$$

**Corollary 4.6.** *A softer bound $C'$ is obtained, which can be expressed by $\hat{\varepsilon}_r(\pi_{\mathcal{D}})$, $\hat{\varepsilon}_m(\pi_{\mathcal{D}})$ and $\hat{\varepsilon}_p(\pi)$ as follows:*

$$C' = \sum_{t=0}^{\Gamma(\pi)-1} \gamma^t \left( 4r_{max}(t+1)\hat{\varepsilon}_p(\pi) + (2 - \frac{\lambda_r}{\alpha_r})\hat{\varepsilon}_r(\pi_{\mathcal{D}}) + 2r_{max}(t+1)\hat{\varepsilon}_m(\pi_{\mathcal{D}}) \right) \quad (12)$$

In previous works, the world model is trained before the policy optimization, with a loss function measuring the mean square error (MSE) between the predictions and true values in the offline dataset: $L = \mathbb{E}_{(s_t,a_t)\sim\pi|\mathcal{M}} [|R(s_t,a_t) - r_\eta(s_t,a_t)|] + \mathbb{E}_{(s_t,a_t)\sim\pi|\mathcal{M}} [D_{TV}(P(s_{t+1}|s_t,a_t)\|P_\theta(s_{t+1}|s_t,a_t))]$. Thus, $\hat{\varepsilon}_r(\pi_{\mathcal{D}})$ and $\hat{\varepsilon}_m(\pi_{\mathcal{D}})$ in equation 12 is minimized. However, $\hat{\varepsilon}_p(\pi)$ is never optimized by the world model, which can cause the distributional shift problem during policy evaluation. As the target policy deviated from the behavior policy, the bound $C'$ grows larger, and the monotonic improvement can't be guaranteed. On the contrary, a fundamental difference in our algorithm is to consider minimizing a tighter bound $C$. We adopt importance sampling to estimate $\hat{\varepsilon}_r(\pi)$ and $\hat{\varepsilon}_r(\pi)$ on offline datasets, and design our loss function based on $L_1 = \mathbb{E}_{(s_t,a_t)\sim\pi_{\mathcal{D}}|\mathcal{M}} \left[ \frac{\pi(s_t,a_t)}{\pi_{\mathcal{D}}(s_t,a_t)} [D_{TV}(R(s_t,a_t)\|r_\eta(s_t,a_t))] \right]$ and $L_2 = \mathbb{E}_{(s_t,a_t)\sim\pi_{\mathcal{D}}|\mathcal{M}} \left[ \frac{\pi(s_t,a_t)}{\pi_{\mathcal{D}}(s_t,a_t)} [D_{TV}(P(s_{t+1}|s_t,a_t)\|P_\theta(s_{t+1}|s_t,a_t))] \right]$. These two loss function are taking over samples drawn from the behavior policy, but with importance sampling re-weighting $\frac{\pi(s_t,a_t)}{\pi_{\mathcal{D}}(s_t,a_t)}$, thus minimizing the return gap $C$ directly.

With this theorem, the monotonic improvement of the true return $J(\pi)$ is guaranteed theoretically when the returns under uncertainty-penalized diffusion world model $\hat{J}_{\theta,\eta}(\pi)$ is improved by more than $C$. However, practically the monotonic improvement could face trouble due to the following limitations:

- **The compounding error**: Though the diffusion model has lower prediction error after training on $\mathcal{D}$ than traditional MLPs, the state transition error in each step accumulates as the compounding error, decreasing long-term planning accuracy.

- **Out of distribution**: While using the diffusion world model for policy evaluation, the action derived from the policy can drive the state out of the distribution represented by $\mathcal{D}$. In that case, the generated state and reward become unstable, causing the overestimation of returns from unknown state-action pairs.

- **Aleatory uncertainty**: In this method, the uncertainty estimator could have trouble to distinguish the aleatory uncertainty and the epistemic uncertainty. Though such a method is competitive to determined MDPs or MDPs with Lipschitz transitions, further study should be considered before moving to stochastic environments.

To handle these, we set $H$ and $\lambda_0$ small enough to limit the accumulating prediction errors, while randomly choosing the initial state from the whole dataset and replay buffer. To avoid overestimating out-of-distribution state-action pairs, we choose proper uncertainty-based threshold and penalty parameters, preventing the policy from visiting these pairs. Finally, to solve the aleatory uncertainty issue, there has been work using a similar idea with a single diffusion model and a Bayesian hyper-network (Chan et al.), which we consider for future work.

## 5 EXPERIMENTS

Our experiments are designed to evaluate: 1. The performance of ADEPT with adaptive diffusion world model and offline RL updates, compared with other SOTA algorithms, including diffusion-based methods. 2. The effectiveness of the proposed importance sampling and uncertainty penalty techniques in ADEPT. We train and test our method on multiple environments and datasets in D4RL (Fu et al., 2020) to show the quantitative results, and further analyze our method with ablation study.

### 5.1 NUMERICAL EVALUATION

In this section, we evaluate our proposed ADEPT algorithm over multiple MuJoCo environments including Locomotion (halfcheetah, walker2d, and hopper) with 4 different datasets (random, medium, medium-replay and medium-expert), Maze2d, AntMaze and Adroit (pen-human, pen-cloned). We

select a number of SOTA algorithms as baselines, including model-free methods TD3+BC (Fujimoto & Gu, 2021), CQL (Kumar et al., 2020), IQL (Kostrikov et al., 2021), model-based methods such as RAMBO (Rigter et al., 2022), MOPO (Yu et al., 2020), COMBO (Yu et al., 2021), and diffusion-based methods as SyntheER (Lu et al., 2023) and Diffuser (Janner et al., 2022), as well as behavior cloning. All experiments are conducted with the same training hyperparameters. The comparison is summarized in Table 1.

| Environment | Dataset | TD3+BC | CQL | IQL | RAMBO | MOPO | COMBO | SynthER | Diffuser | ADEPT(Ours) |
|---|---|---|---|---|---|---|---|---|---|---|
| halfcheetah | | $11.3 \pm 0.8$ | $\mathbf{35.4 \pm 0.9}$ | $12.5 \pm 1.2$ | $33.5 \pm 2.6$ | $\mathbf{35.4 \pm 2.5}$ | $38.8 \pm 3.7$ | $17.2 \pm 3.4$ | $3.6 \pm 2.5$ | $34.5 \pm 1.1$ |
| walker2d | rnd | $0.6 \pm 0.3$ | $7.0 \pm 1.2$ | $5.4 \pm 0.8$ | $0.2 \pm 0.6$ | $\mathbf{13.6 \pm 2.6}$ | $7.0 \pm 3.6$ | $4.2 \pm 0.3$ | $3.5 \pm 1.4$ | $10.3 \pm 2.2$ |
| hopper | | $8.6 \pm 0.3$ | $10.8 \pm 0.1$ | $7.5 \pm 0.2$ | $15.5 \pm 9.4$ | $11.7 \pm 0.4$ | $17.9 \pm 1.4$ | $7.7 \pm 0.1$ | $6.3 \pm 0.8$ | $\mathbf{31.7 \pm 0.9}$ |
| halfcheetah | | $48.1 \pm 0.2$ | $44.4 \pm 0.3$ | $47.4 \pm 0.1$ | $\mathbf{71.0 \pm 3.0}$ | $42.3 \pm 1.6$ | $54.2 \pm 1.5$ | $49.6 \pm 0.3$ | $42.8 \pm 0.3$ | $62.1 \pm 0.5$ |
| walker2d | med | $82.7 \pm 5.5$ | $79.2 \pm 8.3$ | $78.3 \pm 5.4$ | $89.1 \pm 2.7$ | $17.8 \pm 19.3$ | $81.9 \pm 2.8$ | $84.7 \pm 5.5$ | $79.6 \pm 0.6$ | $\mathbf{97.2 \pm 2.5}$ |
| hopper | | $60.4 \pm 4.0$ | $58.0 \pm 10.8$ | $66.3 \pm 6.0$ | $91.2 \pm 16.3$ | $28.0 \pm 12.4$ | $97.2 \pm 2.2$ | $72.0 \pm 4.5$ | $74.3 \pm 1.4$ | $\mathbf{107.7 \pm 1.5}$ |
| halfcheetah | | $44.8 \pm 0.7$ | $46.2 \pm 1.1$ | $44.2 \pm 0.4$ | $\mathbf{67.0 \pm 1.5}$ | $53.1 \pm 2.0$ | $55.1 \pm 1.0$ | $46.6 \pm 0.2$ | $37.7 \pm 0.5$ | $56.8 \pm 1.2$ |
| walker2d | med-rep | $85.6 \pm 4.6$ | $26.7 \pm 2.6$ | $94.7 \pm 8.0$ | $88.5 \pm 4.0$ | $39.0 \pm 9.6$ | $56.0 \pm 8.6$ | $83.3 \pm 5.9$ | $70.6 \pm 1.6$ | $\mathbf{101.5 \pm 1.4}$ |
| hopper | | $64.4 \pm 24.8$ | $48.6 \pm 0.9$ | $73.9 \pm 13.5$ | $97.6 \pm 3.4$ | $67.5 \pm 24.7$ | $89.5 \pm 1.8$ | $\mathbf{103.2 \pm 0.4}$ | $93.6 \pm 0.4$ | $\mathbf{103.4 \pm 3.7}$ |
| halfcheetah | | $90.8 \pm 7.0$ | $62.4 \pm 3.9$ | $86.7 \pm 0.2$ | $79.3 \pm 2.9$ | $63.3 \pm 38.0$ | $90.0 \pm 5.6$ | $93.3 \pm 2.6$ | $88.9 \pm 0.3$ | $\mathbf{94.6 \pm 1.1}$ |
| walker2d | med-exp | $\mathbf{110.0 \pm 0.4}$ | $98.7 \pm 13.1$ | $\mathbf{109.6 \pm 0.6}$ | $63.1 \pm 31.3$ | $44.6 \pm 12.9$ | $103.3 \pm 5.6$ | $\mathbf{111.4 \pm 0.7}$ | $106.9 \pm 0.2$ | $\mathbf{111.5 \pm 1.9}$ |
| hopper | | $101.1 \pm 10.5$ | $\mathbf{111.0 \pm 1.2}$ | $91.5 \pm 6.1$ | $89.5 \pm 11.1$ | $23.7 \pm 6.0$ | $\mathbf{111.1 \pm 2.9}$ | $90.8 \pm 17.9$ | $103.3 \pm 1.3$ | $\mathbf{113.3 \pm 2.3}$ |
| Average | | $59.0 \pm 4.9$ | $52.4 \pm 3.7$ | $59.8 \pm 3.5$ | $65.5 \pm 7.4$ | $36.6 \pm 11.0$ | $66.8 \pm 3.4$ | $63.7 \pm 3.5$ | $59.3 \pm 0.9$ | $\mathbf{77.1 \pm 1.7}$ |

Table 1: The evaluation of ADEPT compared with other SOTA offline RL algorithms, on D4RL MuJoCo environments with random (rnd), medium (med), medium-replay (med-rep) and medium-expert (med-exp) datasets. We show the mean and standard deviation of the performance over 5 different seeds. The statistically significant results are noted in bold.

From our experimental result, ADEPT outperforms the existing SOTA offline RL algorithms in most of the environments, especially in medium dataset. This is consistent with our hypothesis that world model adaptation is more critical when there is a lack of expert demonstrations, and the distributional shift becomes more severe as target policy moves toward optimum. Compared to model-free method which performs poorly due to lack of policy regularization, the diffusion model generated data has significantly improved its performance. This is because the uncertainty penalty and importance-sampled adaption could be viewed as a conservative regularization method. For out-of-distribution situations, the diffusion model generates more relevant state transition results and corresponding pessimistic reward function with uncertainty penalty, which mitigates overestimation.

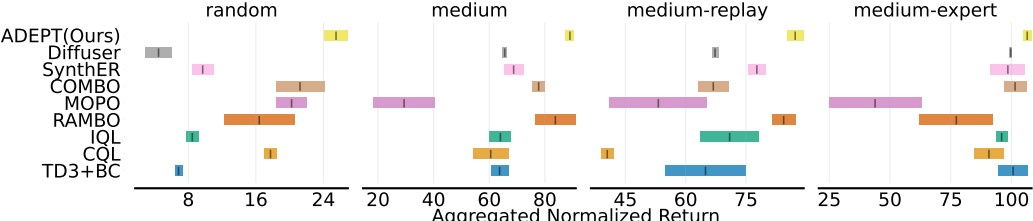

Figure 3: Aggregate performance over D4RL MuJoCo environments. Mean return is marked with standard error highlighted.

Figure 3 shows the aggregate performance over the three environments. Compared with the highest score of other SOTA algorithms, our method gains an average of 20.1% improvement over the SOTA algorithm on random dataset and achieves significant improvement over nearly all baselines on medium and medium-replay datasets, while the improvement is limited in medium-expert dataset. Such a result shows that with a lack of expert demonstration and plenty of sub-optimal data, a closed-loop iterative algorithm for diffusion world model adaptation and offline policy improvement could become more advantageous than using diffusion models as multistep planner like Diffuser (Janner et al., 2022), or generating synthetic dataset one time before training as SynthER (Lu et al., 2023).

## 5.2 ABLATION STUDY

In the ablation study, we intend to validate the necessity of uncertainty estimation and the effectiveness of importance sampling in the close-loop ADEPT algorithm. To accomplish this, we compare our methods under different settings: (1) SAC: Using original SAC with no generated data used; (2) ADEPT w/o IS: Model-based RL with diffusion model trained one-time before training, so no importance sampling technique; (3) ADEPT w/o UE: removing uncertainty estimation and reward penalty from the diffusion model. These three settings show the importance of adaptive diffusion world model with importance sampling and uncertainty estimation, respectively. We also perform additional experiments on the hyperparameters which work as a trade-off between performance and efficiency, combination of the uncertainty-penalized diffusion model and other offline model-free RL algorithms. The results are shown in the Appendix.

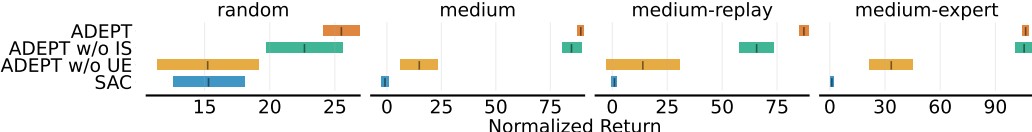

(a) Aggregated results of different behavior policies

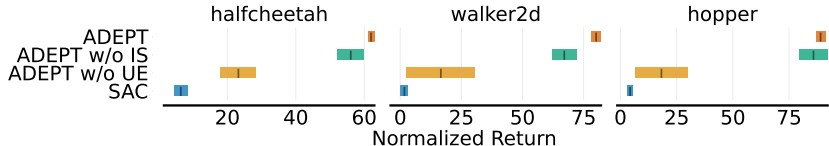

(b) Aggregated results of different environments

Figure 4: Ablation study on key components in ADEPT

Figure 4 shows the aggregated results of the modified methods in MuJoCo environments. With no synthetic data, the original SAC algorithm fails in all the datasets except the random, while its performance in other datasets is even lower than behavior cloning. The use of diffusion world model largely enhances the performance of SAC by providing plausible results of out-of-distribution samples to avoid overestimation. Besides, we noticed that without uncertainty penalty, the training process is unstable and faces severe performance drops. Also, adopting importance sampling for diffusion model adaptation improves the performance of SAC substantially in all but medium and medium-expert dataset. Its limited effects on these datasets are consistent with the hypothesis that samples in the suboptimal dataset are more scattered, so the diffusion world model adaptation could have a higher impact on the generated distribution, aligning the world model with the target policy.

## 6 CONCLUSION

This paper proposes a new model-based offline RL algorithm ADEPT adopting (i) an uncertainty-penalized diffusion world model to directly evaluate and optimize the target policy in offline reinforcement learning and (ii) an importance-sampled world model update to adaptively align the world model with the evolving policy, in a closed-loop operation. These two key components enable ADEPT to significantly reduce the distributional shift problem and avoid reward overestimation under out-of-distribution state and action pairs. Our theoretical analysis of the algorithm provides an upper bound on the return gap and illuminates key factors affecting the learning performance. Experimental results on the D4RL benchmark show that ADEPT significantly improves the total performance and training stability of SAC, and substantially outperforms other state-of-the-art offline RL baselines in almost all tasks. Compared to the best scores of other algorithms, ADEPT achieves an average advantage of 15.4%, with 20.1% on random dataset, and 6.2% on medium dataset. Our work provides important insights into the use of diffusion-based world model in offline RL. Exciting future research could extend uncertainty-penalized diffusion model to a multistep planner, or generalize it to more complicated scenes such as partial observable, multi-agent or stochastic environments.

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

# A  PROOF OF THE THEOREM

We show the complete proof for the return gap bound proposed in the main paper, inspired by Yu et al. (2020), Kidambi et al. (2020), and Janner et al. (2019).

**Lemma A.1.** *For any 2 different joint distribution of states and actions $P^\pi(s, a)$ under $\pi$ and $\mathcal{M}$, and $\hat{P}^{\hat{\pi}}(s, a)$ under $\hat{\pi}$ and $\hat{\mathcal{M}}$, we have*

$$D_{TV}(P^\pi(s,a)\|\hat{P}^{\hat{\pi}}(s,a)) \le D_{TV}(P^\pi(s)\|\hat{P}^{\hat{\pi}}(s)) + \max_s D_{TV}(\pi(a|s)\|\hat{\pi}(a|s)).$$

*Proof.*

$$D_{TV}(P^\pi(s,a)\|\hat{P}^{\hat{\pi}}(s,a)) = \frac{1}{2}\sum_{s,a}|P^\pi(s,a) - \hat{P}^{\hat{\pi}}(s,a)|$$

$$= \frac{1}{2}\sum_{s,a}|P^\pi(s)\pi(a|s) - \hat{P}^{\hat{\pi}}(s,a)\hat{\pi}(a|s)|$$

$$= \frac{1}{2}\sum_{s,a}|P^\pi(s)\pi(a|s) - P^\pi(s)\hat{\pi}(a|s) + P^\pi(s)\hat{\pi}(a|s) - \hat{P}^{\hat{\pi}}(s,a)\hat{\pi}(a|s)|$$

$$\leq \frac{1}{2}\sum_{s,a}P^\pi(s)|\pi(a|s) - \hat{\pi}(a|s)| + \frac{1}{2}\sum_{s,a}|P^\pi(s) - \hat{P}^{\hat{\pi}}(s)|\hat{\pi}(a|s)$$

$$= \mathbb{E}_{s\sim P^\pi}\left[D_{TV}(\pi(a|s)\|\hat{\pi}(a|s))\right] + \frac{1}{2}\sum_s|P^\pi(s) - \hat{P}^{\hat{\pi}}(s)|$$

$$= \mathbb{E}_{s\sim P^\pi}\left[D_{TV}(\pi(a|s)\|\hat{\pi}(a|s))\right] + D_{TV}\left(P^\pi(s)\|\hat{P}^{\hat{\pi}}(s)\right)$$

$$\leq D_{TV}\left(P^\pi(s)\|\hat{P}^{\hat{\pi}}(s)\right) + \max_s D_{TV}(\pi(a|s)\|\hat{\pi}(a|s)).$$

$\square$

To be noted that this equation can be extended to conditional probabilities, such as:

$$D_{TV}(P^\pi(s',a'|s,a)\|\hat{P}^{\hat{\pi}}(s',a'|s,a)) \leq D_{TV}(P(s'|s,a)\|\hat{P}(s'|s,a)) + \max_{s'}D_{TV}(\pi(a'|s')\|\hat{\pi}(a'|s')).$$

Here with a little abuse of notation, we simplify $P_t^\pi(s',a'|s_{t-1}=s,a_{t-1}=a)$ as $P^\pi(s',a'|s,a)$, and $\hat{P}_t^{\hat{\pi}}(s',a'|s_{t-1}=s,a_{t-1}=a)$ as $\hat{P}^{\hat{\pi}}(s',a'|s,a)$, which is also used in the rest of the paper. Note that $t$ can be omitted since we are considering a Markov process.

**Lemma A.2.** *For two joint distributions of states and actions at time step $t$ noted as $P_t^\pi(s,a)$ under $\pi$ and $\mathcal{M}$, and $\hat{P}_t^{\hat{\pi}}(s,a)$ under $\hat{\pi}$ and $\hat{\mathcal{M}}$, given the same initial state distribution as $P_0^\pi(s) = \hat{P}_0^{\hat{\pi}}(s) = \mu_0(s), \forall s$, and there exists $\delta$ s.t.*

$$\max_t \mathbb{E}_{(s,a)\sim P_t^\pi} D_{TV}(P^\pi(s',a'|s,a)\|\hat{P}^{\hat{\pi}}(s',a'|s,a)) \leq \delta. \tag{13}$$

*Then we have*

$$D_{TV}(P_t^\pi(s,a)\|\hat{P}_t^{\hat{\pi}}(s,a)) \leq t\delta + D_{TV}(P_0^\pi(s,a)\|\hat{P}_0^{\hat{\pi}}(s,a)). \tag{14}$$

*Proof.*

$$\left|P_t^\pi(s',a') - \hat{P}_t^{\hat{\pi}}(s',a')\right| = \left|\sum_{s,a}\left(P_{t-1}^\pi(s,a)P^\pi(s',a'|s,a) - \hat{P}_{t-1}^{\hat{\pi}}(s,a)\hat{P}^{\hat{\pi}}(s',a'|s,a)\right)\right|$$

$$\leq \sum_{s,a}\left|\left(P_{t-1}^\pi(s,a)P^\pi(s',a'|s,a) - \hat{P}_{t-1}^{\hat{\pi}}(s,a)\hat{P}^{\hat{\pi}}(s',a'|s,a)\right)\right|$$

$$= \sum_{s,a}\left|\left(P_{t-1}^\pi(s,a)\left(P^\pi(s',a'|s,a) - \hat{P}^{\hat{\pi}}(s',a'|s,a)\right)\right) + \hat{P}^{\hat{\pi}}(s',a'|s.a)\left(P_{t-1}^\pi(s,a) - \hat{P}_{t-1}^{\hat{\pi}}(s,a)\right)\right|$$

$$\leq \sum_{s,a}\left(P_{t-1}^\pi(s,a)\left|P^\pi(s',a'|s,a) - \hat{P}^{\hat{\pi}}(s',a'|s.a)\right| + \hat{P}^{\hat{\pi}}(s',a'|s,a)\left|P_{t-1}^\pi(s,a) - \hat{P}_{t-1}^{\hat{\pi}}(s,a)\right|\right)$$

$$= \mathbb{E}_{(s,a)\sim P_{t-1}^\pi}\left|P^\pi(s',a'|s,a) - \hat{P}^{\hat{\pi}}(s',a'|s,a)\right| + \sum_{s,a}\hat{P}^{\hat{\pi}}(s',a'|s,a)\left|P_{t-1}^\pi(s,a) - \hat{P}_{t-1}^{\hat{\pi}}(s,a)\right|.$$

Therefore, we have:

$$D_{TV}(P_t^\pi(s,a)\|\hat{P}_t^{\hat{\pi}}(s,a)) = \frac{1}{2}\sum_{s',a'}\left|P_t^\pi(s',a') - \hat{P}_t^{\hat{\pi}}(s',a')\right|$$

$$\leq \frac{1}{2}\sum_{s',a'}\left(\mathbb{E}_{s,a\sim P_{t-1}^\pi}\left|P^\pi(s',a'|s,a) - \hat{P}^{\hat{\pi}}(s',a'|s,a)\right| + \sum_{s,a}\hat{P}^{\hat{\pi}}(s',a'|s,a)\left|P_{t-1}^\pi(s,a) - \hat{P}_{t-1}^{\hat{\pi}}(s,a)\right|\right)$$

$$= \mathbb{E}_{(s,a)\sim P_{t-1}^\pi}D_{TV}(P^\pi(s',a'|s,a)\|\hat{P}^{\hat{\pi}}(s',a'|s,a)) + \frac{1}{2}\sum_{s',a'}\sum_{s,a}\hat{P}^{\hat{\pi}}(s',a'|s,a)\left|P_{t-1}^\pi(s,a) - \hat{P}_{t-1}^{\hat{\pi}}(s,a)\right|$$

$$\leq \delta + \frac{1}{2}\sum_{s,a}\left|P_{t-1}^\pi(s,a) - \hat{P}_{t-1}^{\hat{\pi}}(s,a)\right|$$

$$= \delta + D_{TV}\left(P_{t-1}^\pi(s,a)\|\hat{P}_{t-1}^{\hat{\pi}}(s,a)\right)$$

$$\leq t\delta + D_{TV}(P_0^\pi(s,a)\|\hat{P}_0^{\hat{\pi}}(s,a)).$$

Since the initial state distribution is the same for the two environmental model, *i.e.*, $P_0^\pi(s) = \hat{P}_0^{\hat{\pi}}(s) = \mu_0(s), \forall s$, then by applying Lemma A.1 we have:

$$D_{TV}(P_t^\pi(s,a)\|\hat{P}_t^{\hat{\pi}}(s,a)) \leq t\delta + D_{TV}(P_0^\pi(s)\|\hat{P}_0^{\hat{\pi}}(s)) + \max_s D_{TV}(\pi(a|s)\|\hat{\pi}(a|s))$$

$$= t\delta + \max_s D_{TV}(\pi(a|s)\|\hat{\pi}(a|s)).$$

$\square$

**Definition A.3.** We define $\Gamma(\pi)$ as the expected time step that the policy on the diffusion world model visits an uncertain state and action pair with a discrepancy larger than $\lambda_0$:

$$\Gamma(\pi) = \mathbb{E}_{(s_t,a_t)\sim\pi|\hat{\mathcal{M}}}\left[\min\{t|d_\theta(s_t,a_t) \geq \lambda_0\}\right] \tag{15}$$

**Lemma A.4.** *We define $r_{max} = \max_{s,a} R(s,a)$. For any $\pi$ under $\mathcal{M}$ and $\hat{\pi}$ under $\hat{\mathcal{M}}_{\theta,\eta}$, satisfying*

$$\max_t \mathbb{E}_{(s_t,a_t)\sim\pi|\mathcal{M}}\left[\frac{1}{2}|R(s_t,a_t) - r_\eta(s_t,a_t)|\right] \leq \hat{\varepsilon}_r(\pi). \tag{16}$$

*we have*

$$J(\pi) - \gamma^{\Gamma(\hat{\pi})}J_{\Gamma(\hat{\pi})}(\pi) - \hat{J}_{\theta,\eta}(\hat{\pi}) \geq -\sum_{t=0}^{\Gamma(\hat{\pi})-1}\gamma^t\left((2 - \frac{\lambda_r}{\alpha_r})\hat{\varepsilon}_r(\pi) + 2r_{max}D_{TV}(P_t^\pi(s,a)\|\hat{P}_t^{\hat{\pi}}(s,a))\right). \tag{17}$$

Here, $J_{\Gamma(\hat{\pi})}(\pi) = \sum_{t=\Gamma(\hat{\pi})}^T \gamma^{t-\Gamma(\hat{\pi})}\sum_{s,a}P_t^\pi(s,a)R(s,a)$ is defined as the expectation of returns starting from the distribution of $P_{\Gamma(\pi)}^\pi(s)$.

*Proof.*

$$J(\pi) - \gamma^{\Gamma(\hat{\pi})} J_{\Gamma(\hat{\pi})}(\pi) - \hat{J}_{\theta,\eta}(\pi) = \sum_{t=0}^{\Gamma(\hat{\pi})-1} \gamma^t \sum_{s,a} \left( P_t^\pi(s,a) R(s,a) - \hat{P}_t^{\hat{\pi}}(s,a) \hat{r}(s,a) \right)$$

$$= \sum_{t=0}^{\Gamma(\hat{\pi})-1} \gamma^t \sum_{s,a} \left( P_t^\pi(s,a) R(s,a) - \hat{P}_t^{\hat{\pi}}(s,a) r_\eta(s,a) + \lambda_r \hat{P}_t^{\hat{\pi}}(s,a) d_\theta(s,a) \right)$$

$$= \sum_{t=0}^{\Gamma(\hat{\pi})-1} \gamma^t \left( \sum_{s,a} \left[ P_t^\pi(s,a) \left( R(s,a) - r_\eta(s,a) \right) + r_\eta(s,a) \left( P_t^\pi(s,a) - \hat{P}_t^{\hat{\pi}}(s,a) \right) + \lambda_r P_t^\pi(s,a) d_\theta(s,a) \right] \right)$$

$$\geq \sum_{t=0}^{\Gamma(\hat{\pi})-1} \gamma^t \left( -\mathbb{E}_{(s_t,a_t)\sim\pi|\mathcal{M}} [|R(s_t,a_t) - r_\eta(s_t,a_t)|] + \sum_{s,a} \left[ r_\eta(s,a) \left( P_t^\pi(s,a) - \hat{P}_t^{\hat{\pi}}(s,a) \right) + \lambda_r P_t^\pi(s,a) d_\theta(s,a) \right] \right)$$

$$\geq \sum_{t=0}^{\Gamma(\hat{\pi})-1} \gamma^t \left( -2\hat{\varepsilon}_r(\pi) + \sum_{s,a} \left[ r_\eta(s,a) \left( P_t^\pi(s,a) - \hat{P}_t^{\hat{\pi}}(s,a) \right) + \lambda_r P_t^\pi(s,a) d_\theta(s,a) \right] \right) \quad \cdots \text{Definition of } \hat{\varepsilon}_r(\pi)$$

$$\geq \sum_{t=0}^{\Gamma(\hat{\pi})-1} \gamma^t \left( -2\hat{\varepsilon}_r(\pi) - r_{max} \sum_{s,a} \left| P_t^\pi(s,a) - \hat{P}_t^{\hat{\pi}}(s,a) \right| + \lambda_r \mathbb{E}_{(s_t,a_t)\sim\pi|\mathcal{M}} [d_\theta(s,a)] \right)$$

$$\geq -\sum_{t=0}^{\Gamma(\hat{\pi})-1} \gamma^t \left( (2 - \frac{\lambda_r}{\alpha_r}) \hat{\varepsilon}_r(\pi) + 2 r_{max} D_{TV}(P_t^\pi(s,a) \| \hat{P}_t^{\hat{\pi}}(s,a)) \right). \quad \cdots \text{Assumption 4.4}$$

Besides, if $\pi$ and $\hat{\pi}$ are under the same model, then there's no reward prediction error in the bound, and we have:

$$(J(\pi) - \gamma^{\Gamma(\pi)} J_{\Gamma(\pi)}(\pi)) - (J(\hat{\pi}) - \gamma^{\Gamma(\pi)} J_{\Gamma(\pi)}(\hat{\pi})) \geq -2 r_{max} \sum_{t=0}^{\Gamma(\pi)-1} \gamma^t D_{TV}(P_t^\pi(s,a) \| P_t^{\hat{\pi}}(s,a)),$$

$$\hat{J}_{\theta,\eta}(\pi) - \hat{J}_{\theta,\eta}(\hat{\pi}) \geq -2 r_{max} \sum_{t=0}^{\Gamma(\pi)-1} \gamma^t D_{TV}(\hat{P}_t^\pi(s,a) \| \hat{P}_t^{\hat{\pi}}(s,a)).$$

$\square$

**Theorem A.5.** *Given $\hat{\varepsilon}_r(\pi)$, $\hat{\varepsilon}_m(\pi)$ with the Definition 4.1 and 4.2, the bound $C$ between the true return $J$ and the ADEPT model return $\hat{J}_{\theta,\eta}$ under the same target policy $\pi$ can be obtained:*

$$C = \sum_{t=0}^{\Gamma(\pi)-1} \gamma^t \left( (2 - \frac{\lambda_r}{\alpha_r}) \hat{\varepsilon}_r(\pi) + 2 r_{max}(t+1) \hat{\varepsilon}_m(\pi) \right)$$

*Proof.* By applying Lemma A.4 we can get:

$$J(\pi) - \gamma^{\Gamma(\pi)} J_{\Gamma(\pi)}(\pi) - \hat{J}_{\theta,\eta}(\pi) \geq - \sum_{t=0}^{\Gamma(\pi)-1} \gamma^t \left( (2 - \frac{\lambda_r}{\alpha_r}) \hat{\varepsilon}_r(\pi) + 2 r_{max} D_{TV}(P_t^\pi(s,a) \| \hat{P}_t^\pi(s,a)) \right).$$

To analyze the last term, we first consider the condition of Lemma A.2. By using Lemma A.1 we have:

$$\max_t \mathbb{E}_{(s,a)\sim P_t^\pi} D_{TV}(P^\pi(s',a'|s,a) \| \hat{P}^\pi(s',a'|s,a))$$

$$\leq \max_t \mathbb{E}_{(s,a)\sim P_t^\pi} D_{TV}(P(s'|s,a) \| \hat{P}(s'|s,a)) + \max_s D_{TV}(\pi(a|s) \| \pi(a|s))$$

$$= \max_t \mathbb{E}_{(s,a)\sim P_t^{\pi^D}} D_{TV}(P(s'|s,a) \| \hat{P}(s'|s,a))$$

$$\leq \hat{\varepsilon}_m(\pi).$$

Next, we apply Lemma A.2 with this result by replacing $\delta$ with $\hat{\varepsilon}_m(\pi)$ and get:

$$J(\pi) - \gamma^{\Gamma(\pi)} J_{\Gamma(\pi)}(\pi) \geq \hat{J}_{\theta,\eta}(\pi) - \sum_{t=0}^{\Gamma(\pi)-1} \gamma^t \left( (2 - \frac{\lambda_r}{\alpha_r}) \hat{\varepsilon}_r(\pi) + 2 r_{max}(t+1) \hat{\varepsilon}_m(\pi) \right)$$

$\square$

**Corollary A.6.** *In traditional algorithms where importance sampling is not adopted, the world model can only be trained by optimizing $\hat{\varepsilon}_r(\pi_{\mathcal{D}})$ and $\hat{\varepsilon}_m(\pi_{\mathcal{D}})$. Thus, a softer bound $C'$ is obtained to be expressed by $\hat{\varepsilon}_r(\pi_{\mathcal{D}})$, $\hat{\varepsilon}_m(\pi_{\mathcal{D}})$ and $\hat{\varepsilon}_p(\pi)$ as follows:*

$$C' = \sum_{t=0}^{\Gamma(\pi)-1} \gamma^t \left( 4r_{max}(t+1)\hat{\varepsilon}_p(\pi) + (2 - \frac{\lambda_r}{\alpha_r})\hat{\varepsilon}_r(\pi_{\mathcal{D}}) + 2r_{max}(t+1)\hat{\varepsilon}_m(\pi_{\mathcal{D}}) \right)$$

*Proof.* We denote $\pi_{\mathcal{D}}$ as the policy collecting the trajectories in the diffusion world model. The return gap could be separated into:

$$J(\pi) - \gamma^{\Gamma(\pi)}J_{\Gamma(\pi)}(\pi) - \hat{J}_{\theta,\eta}(\pi) = [(J(\pi) - \gamma^{\Gamma(\pi)}J_{\Gamma(\pi)}(\pi)) - (J(\pi_{\mathcal{D}}) - \gamma^{\Gamma(\pi)}J_{\Gamma(\pi)}(\pi_{\mathcal{D}}))] +$$

$$(J(\pi_{\mathcal{D}}) - \gamma^{\Gamma(\pi)}J_{\Gamma(\pi)}(\pi_{\mathcal{D}}) - \hat{J}_{\theta,\eta}(\pi_{\mathcal{D}})) + (\hat{J}_{\theta,\eta}(\pi_{\mathcal{D}}) - \hat{J}_{\theta,\eta}(\pi)).$$

Next we analyze these three parts one by one. By applying Lemma A.4, we have:

$$(J(\pi) - \gamma^{\Gamma(\pi)}J_{\Gamma(\pi)}(\pi)) - (J(\pi_{\mathcal{D}}) - \gamma^{\Gamma(\pi)}J_{\Gamma(\pi)}(\pi_{\mathcal{D}})) \geq -2r_{max} \sum_{t=0}^{\Gamma(\pi)-1} \gamma^t \left( D_{TV}(P_t^{\pi_{\mathcal{D}}}(s,a) \| P_t^{\pi}(s,a)) \right).$$

Considering the condition in Lemma A.2, according to Lemma A.1, we can bound it by:

$$\max_t \mathbb{E}_{(s,a) \sim P_t^{\pi_{\mathcal{D}}}} D_{TV}(P^{\pi_{\mathcal{D}}}(s',a'|s,a) \| P^{\pi}(s',a'|s,a))$$

$$\leq \max_t \mathbb{E}_{(s,a) \sim P_t^{\pi_{\mathcal{D}}}} D_{TV}(P(s'|s,a) \| P(s'|s,a)) + \max_s D_{TV}(\pi_{\mathcal{D}}(a|s) \| \pi(a|s))$$

$$= \max_s D_{TV}(\pi_{\mathcal{D}}(a|s) \| \pi(a|s))$$

$$\leq \hat{\varepsilon}_p(\pi).$$

Therefore, we replace $\delta$ with $\hat{\varepsilon}_p(\pi)$ in the condition of Lemma A.2 and get:

$$(J(\pi) - \gamma^{\Gamma(\pi)}J_{\Gamma(\pi)}(\pi)) - (J(\pi_{\mathcal{D}}) - \gamma^{\Gamma(\pi)}J_{\Gamma(\pi)}(\pi_{\mathcal{D}})) \geq -2r_{max} \sum_{t=0}^{\Gamma(\pi)-1} \gamma^t D_{TV}(P_t^{\pi}(s,a) \| P_t^{\pi_{\mathcal{D}}}(s,a))$$

$$\geq -2r_{max} \sum_{t=0}^{\Gamma(\pi)-1} \gamma^t(t+1)\hat{\varepsilon}_p(\pi).$$

The third term could be analyzed similarly, and we get:

$$\hat{J}_{\theta,\eta}(\pi_{\mathcal{D}}) - \hat{J}_{\theta,\eta}(\pi) \geq -2r_{max} \sum_{t=0}^{\Gamma(\pi)-1} \gamma^t D_{TV}(\hat{P}_t^{\pi_{\mathcal{D}}}(s,a) \| \hat{P}_t^{\pi}(s,a))$$

$$\geq -2r_{max} \sum_{t=0}^{\Gamma(\pi)-1} \gamma^t(t+1)\hat{\varepsilon}_p(\pi).$$

For the second part, by applying Lemma A.4 we can get:

$$J(\pi_{\mathcal{D}}) - \gamma^{\Gamma(\pi)}J_{\Gamma(\pi)}(\pi_{\mathcal{D}}) - \hat{J}_{\theta,\eta}(\pi_{\mathcal{D}}) \geq - \sum_{t=0}^{\Gamma(\pi)-1} \gamma^t \left( (2 - \frac{\lambda_r}{\alpha_r})\hat{\varepsilon}_r(\pi_{\mathcal{D}}) + 2r_{max}D_{TV}(P_t^{\pi_{\mathcal{D}}}(s,a) \| \hat{P}_t^{\pi_{\mathcal{D}}}(s,a)) \right).$$

Similarly, we analyze the last term by using Lemma A.1:

$$\max_t \mathbb{E}_{(s,a) \sim P_t^{\pi_{\mathcal{D}}}} D_{TV}(P^{\pi_{\mathcal{D}}}(s',a'|s,a) \| \hat{P}^{\pi_{\mathcal{D}}}(s',a'|s,a))$$

$$\leq \max_t \mathbb{E}_{(s,a) \sim P_t^{\pi_{\mathcal{D}}}} D_{TV}(P(s'|s,a) \| \hat{P}(s'|s,a)) + \max_s D_{TV}(\pi_{\mathcal{D}}(a|s) \| \pi_{\mathcal{D}}(a|s))$$

$$= \max_t \mathbb{E}_{(s,a) \sim P_t^{\pi_{\mathcal{D}}}} D_{TV}(P(s'|s,a) \| \hat{P}(s'|s,a))$$

$$\leq \hat{\varepsilon}_m(\pi_{\mathcal{D}}).$$

By replacing $\delta$ with $\hat{\varepsilon}_m(\pi_{\mathcal{D}})$ in Lemma A.2 we get:

$$J(\pi_{\mathcal{D}}) - \gamma^{\Gamma(\pi)} J_{\Gamma(\pi)}(\pi_{\mathcal{D}}) - \hat{J}_{\theta,\eta}(\pi_{\mathcal{D}}) \geq - \sum_{t=0}^{\Gamma(\pi)-1} \gamma^t \left( (2 - \frac{\lambda_r}{\alpha_r})\hat{\varepsilon}_r(\pi_{\mathcal{D}}) + 2r_{max}(t+1)\hat{\varepsilon}_m(\pi_{\mathcal{D}}) \right).$$

Finally, we summed the bounds of all three parts and get:

$$J(\pi) - \gamma^{\Gamma(\pi)} J_{\Gamma(\pi)}(\pi) \geq \hat{J}_{\theta,\eta}(\pi)$$
$$- \sum_{t=0}^{\Gamma(\pi)-1} \gamma^t \left( 4r_{max}(t+1)\hat{\varepsilon}_p(\pi) + (2 - \frac{\lambda_r}{\alpha_r})\hat{\varepsilon}_r(\pi_{\mathcal{D}}) + 2r_{max}(t+1)\hat{\varepsilon}_m(\pi_{\mathcal{D}}) \right)$$

$\square$

## B    EXPERIMENT DETAILS

We use D4RL (Fu et al., 2020) datasets for evaluation, and the code could be found at `https://github.com/Farama-Foundation/D4RL`. These datasets are licensed under the Creative Commons Attribution 4.0 License (CC BY), and their code is licensed under the Apache 2.0 License.

### B.1    BASELINES

We select a number of SOTA baselines algorithms, including model-free methods IQL(Kostrikov et al., 2021), SAC (Haarnoja et al., 2018), TD3+BC (Fujimoto & Gu, 2021), CQL (Kumar et al., 2020), model-based methods such as RAMBO (Rigter et al., 2022), MOPO (Yu et al., 2020), COMBO (Yu et al., 2021), and diffusion-based methods as SyntheER (Lu et al., 2023) and Diffuser (Janner et al., 2022). We run the SAC code for evaluation to get the result, while other results on D4RL dataset are obtained from the original paper of each method. Specially, Diffuser doesn't report their results in random dataset, therefore we run its code from `https://github.com/jannerm/diffuser` for evaluation.

### B.2    COMPUTATIONAL RESOURCES AND COSTS

All of the experiments in this paper are conducted on a server with an AMD EPYC 7513 32-Core Processor CPU and an NVIDIA RTX A6000 GPU. The training of the diffusion model costs approximately 3 hours for 1M gradient steps. The offline training with importance-sampling adaptation costs nearly 15 hours for 1M gradient steps.

### B.3    HYPERPARAMETER AND ARCHITECTURAL DETAILS

In this work we represent the noise prediction model $\epsilon_\theta$ with a residual MLP, while other models including reward prediction model $r_\eta$, actor and critic are traditional MLP with ReLu as the activation function. We show the hyperparameters used in the training process in Table 2 and 3. These hyperparameters are shared in all of the environments.

## C    ADDITIONAL EXPERIMENTAL RESULTS

### C.1    TRADE-OFF BETWEEN PERFORMANCE AND EFFICIENCY

In ADEPT, higher values of the denoising steps $K$ and larger network size of $\epsilon_\theta$ generally have higher accuracy and robustness on the state prediction, leading to a better performance. However, increasing these hyperparameters will significantly extend the denoising process of DDPM. To keep a reasonable balance between performance and efficiency, we conducted additional experiments to select these hyperparameters and present the results in Table 4.

| Parameter | Value |
|---|---|
| denoising steps $K$ | 5 |
| train batch size $B_m$ | 1024 |
| learning rate | $1 \times 10^{-4}$ |
| optimizer | Adam |
| hidden dimension of $\epsilon_\theta$ | 1024 |
| depth of $\epsilon_\theta$ | 6 |
| hidden dimension of $r_\eta$ | 256 |
| depth of $r_\eta$ | 2 |
| timestep embedding dimension | 32 |
| model training steps | $1 \times 10^6$ |
| Horizon $H$ | 5 |
| temperature | 0.5 |
| uncertainty penalty coefficient $\lambda_r$ | 50 |
| uncertainty threshold $\lambda_0$ | 0.01 |
| uncertainty sampling times $N_d$ | 5 |

Table 2: Diffusion Training Hyperparameters

| Parameter | Value |
|---|---|
| $\gamma$ | 0.99 |
| actor learning rate | $3 \times 10^{-4}$ |
| critic learning rate | $1 \times 10^{-4}$ |
| train batch $B_p$ | 256 |
| replay buffer size | $1.25 \times 10^6$ |
| evaluation steps per epoch $N_e$ | 50000 |
| gradient steps per epoch | 1000 |
| training epochs | 2000 |
| optimizer | Adam |
| network hidden dimension | 256 |
| network depth | 2 |
| soft target updata rate | $5 \times 10^{-3}$ |

Table 3: Offline RL Hyperparameters

## C.2 TRADE-OFF BETWEEN EXPLORATION AND EXPLOITATION

Figure 5 shows the correlation between the discrepancy as the uncertainty estimation and the actual state prediction error in the halfcheetah dataset collected by four different behavior policies. Data

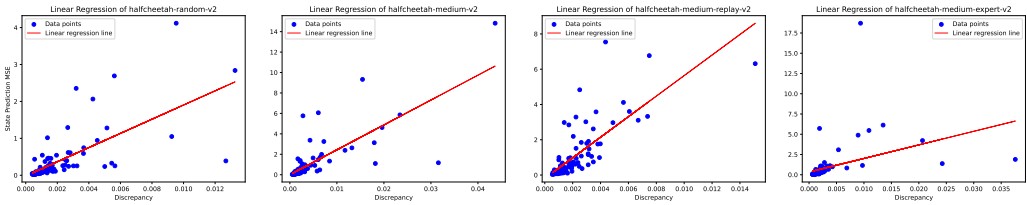

Figure 5: Linear regression of discrepancy and state prediction error in halfcheetah dataset

| Network Width | $K$ | State Prediction MSE | Total Training Time |
|---|---|---|---|
| | 2 | 9.39 | 10.5h |
| 256 | 5 | 4.88 | 12.6h |
| | 10 | 5.04 | 15.1h |
| | 20 | 5.22 | 20.4h |
| | 2 | 9.01 | 12.8h |
| 512 | 5 | 3.56 | 14.4h |
| | 10 | 4.13 | 17.9h |
| | 20 | 4.56 | 25.0h |
| | 2 | 8.61 | 15.6h |
| 1024 | 5 | 3.02 | 18.1h |
| | 10 | 2.79 | 22.9h |
| | 20 | 3.70 | 31.4h |

Table 4: Hyperparameter search for $K$ and network width on halfcheetah-medium dataset

with larger discrepancy tends to have higher prediction error except for few outliers. Besides, we notice that the distribution varies with different behavior policies. For example, the average prediction error and discrepancy are higher in medium dataset than in random dataset, since the distribution of state-action dataset is largely narrowed under a less stochastic policy.

Based on the above, we find that the selection of uncertainty penalty coefficient $\lambda_r$ and uncertainty threshold $\lambda_0$ determines the tolerance of exploring out-of-distribution state-action pairs under the world model and affects the final performance. With higher $\lambda_r$ and lower $\lambda_0$, the target policy will be more conservative and tend to clone behaviors in the dataset, and its value function will predict out-of-distribution state-action pairs with lower reward, which increases the training stability. On the contrary, with low $\lambda_r$ and high $\lambda_0$, the target policy is encouraged to explore the environment and find potential trajectories with higher cumulative rewards. We conducted additional experiments on hopper-random and hopper-medium environments to verify this and show the result in Table 5. The results show that diffusion model trained on a dataset collected by a more stochastic policy has lower average state prediction error. To get best performance, a higher $\lambda_0$ and lower $\lambda_r$ should be selected than those in medium dataset.

| Environment | average state prediction error | $\lambda_0$ | $\lambda_r$ | Normalized Return |
|---|---|---|---|---|
| | | 0.005 | 100 | $17.4 \pm 10.2$ |
| hopper-random | 0.062 | 0.01 | 50 | $31.7 \pm 0.9$ |
| | | 0.015 | 25 | $34.6 \pm 2.8$ |
| | | $\infty$ | 0 | $11.3 \pm 8.5$ |
| | | 0.005 | 100 | $80.6 \pm 2.7$ |
| hopper-medium | 0.137 | 0.01 | 50 | $107.7 \pm 1.5$ |
| | | 0.015 | 25 | $52.9 \pm 6.4$ |
| | | $\infty$ | 0 | $0.8 \pm 0.5$ |

Table 5: The performance of different values of $\lambda_0$ and $\lambda_r$ under two datasets

## C.3 COMBINATION OF ADEPT AND OTHER MODEL-FREE OFFLINE RL ALGORITHMS

The uncertainty-penalized diffusion model used in ADEPT is also compatible with other model-free offline RL algorithms. We further show the additional experimental results in Table 6. The results show that adding extra regularization methods into ADEPT didn't improve much on its performance, since the uncertainty penalized diffusion model and importance-sampling adaption have already worked as a method to avoid distributional shift and overestimation problems. Therefore to simplify the implementation, we choose the original SAC algorithm as the backbone offline RL algorithm in the main text.

| Environment | Dataset | ADEPT+ | | | |
|---|---|---|---|---|---|
| | | SAC (Ours) | IQL | CQL | TD3+BC |
| halfcheetah | | **34.5 ± 1.1** | 33.3 ± 0.7 | 32.4 ± 3.4 | 30.1 ± 3.0 |
| walker2d | random | 10.3 ± 2.2 | 9.5 ± 4.7 | 11.3 ± 2.5 | **14.9 ± 3.9** |
| hopper | | 31.7 ± 0.9 | **34.4 ± 1.8** | 30.7 ± 5.6 | 32.0 ± 1.8 |
| halfcheetah | | 62.1 ± 0.5 | 55.4 ± 3.0 | 56.9 ± 3.3 | **62.3 ± 2.7** |
| walker2d | medium | 97.2 ± 2.5 | 97.0 ± 6.3 | 94.6 ± 5.9 | **100.1 ± 4.6** |
| hopper | | **107.7 ± 1.5** | 69.6 ± 3.5 | 47.1 ± 9.4 | 97.5 ± 3.9 |
| halfcheetah | | 56.8 ± 2.3 | 53.2 ± 1.2 | **59.3 ± 2.6** | 52.5 ± 4.1 |
| walker2d | med-rep | 101.5 ± 1.4 | **105.3 ± 2.0** | 81.7 ± 0.7 | 103.6 ± 2.2 |
| hopper | | **103.4 ± 3.7** | 97.6 ± 4.1 | 99.2 ± 1.5 | 101.3 ± 2.7 |
| halfcheetah | | 94.6 ± 1.1 | 87.1 ± 4.7 | **96.8 ± 0.6** | 87.3 ± 1.8 |
| walker2d | med-exp | **111.5 ± 1.9** | 110.0 ± 1.1 | 108.1 ± 0.2 | 103.8 ± 4.5 |
| hopper | | **113.3 ± 2.3** | 111.8 ± 1.3 | 112.4 ± 2.2 | 109.3 ± 0.8 |
| Average | | **77.1** | 72.0 | 70.9 | 74.6 |

Table 6: The evaluation of ADEPT combined with offline RL algorithms on locomotion environments with random, medium, medium-replay(med-rep) and medium-expert(med-exp) datasets. We show the mean and standard deviation of the performance over 5 different seeds. The maximum average returns in each task are noted in bold.

## C.4   RESULTS ON OTHER D4RL EXPERIMENTS

| Environment | BC | SAC | CQL | IQL | ADEPT(Ours) |
|---|---|---|---|---|---|
| maze2d-umaze | 0.4 | **62.7** | 5.7 | 42.1 | **65.2 ± 10.9** |
| maze2d-medium | 0.8 | 21.3 | 5.0 | **34.9** | **33.1 ± 9.4** |
| antmaze-umaze | 55.3 | 0.0 | 74.0 | **87.5** | **88.1 ± 8.6** |
| pen-human-v1 | 25.8 | 6.3 | 37.5 | **71.5** | 69.4 ± 11.3 |
| pen-cloned-v1 | 38.3 | 23.5 | 39.2 | 47.5 | **52.8 ± 8.0** |
| average | 24.1 | 22.8 | 32.3 | **56.7** | **61.7 ± 9.6** |

Table 7: The evaluation of ADEPT on other D4RL environments. We show the mean and standard deviation of the performance over 5 different seeds. The statistically significant returns in each task are noted in bold. Note that the standard deviation is usually large due to the sparse reward function.

We evaluate ADEPT on extra environments in the D4RL benchmark, shown in Table **??**. Consistent performance is again observed, showing the strong generalization capabilities of ADEPT in different environments. We note that since maze2d and antmaze are goal-guided tasks with near-optimal demonstrations in the datasets, the improvement is not as significant as the locomotion tasks where only suboptimal demonstrations are available.

## C.5   ABLATION STUDY ON REWARD PREDICTION MODEL

| Environment | $r(s_t, a_t, \hat{s}_{t+1})$ | $r(s_t, a_t)$ |
|---|---|---|
| halfcheetah-medium | 0.067 ± 0.012 | 0.098 ± 0.011 |
| halfcheetah-medium-replay | 0.093 ± 0.015 | 0.135 ± 0.011 |
| halfcheetah-medium-expert | 0.098 ± 0.009 | 0.126 ± 0.018 |
| hopper-medium | 0.008 ± 0.001 | 0.013 ± 0.002 |
| hopper-medium-replay | 0.008 ± 0.001 | 0.009 ± 0.001 |
| hopper-medium-expert | 0.006 ± 0.001 | 0.010 ± 0.002 |
| walker2d-medium | 0.071 ± 0.005 | 0.084 ± 0.004 |
| walker2d-medium-replay | 0.060 ± 0.008 | 0.076 ± 0.006 |
| walker2d-medium-expert | 0.064 ± 0.004 | 0.076 ± 0.008 |

Table 8: The ablation study of using $\hat{s}_{t+1}$ as the input of the reward model.

Based on these results, we found that introducing $\hat{s}_{t+1}$ into the reward model could largely reduce the prediction error in all datasets by 10% to 40%. A possible explanation for this is that in most environments, including MuJoCo locomotion tasks, the true reward functions are defined directly with $s_t$ and $s_{t+1}$. One common example is defining the moving distance between two states as the reward. Therefore, by introducing $s_{t+1}$ in the input, the trained reward model becomes more similar to the true reward function, thus has more generalizing capability.

## C.6 MORE EXPERIMENTAL RESULTS COMPARED WITH RENCENT DIFFUSION-BASED ALGORITHMS

| Environment | PGD | DWM | ADEPT |
|---|---|---|---|
| halfcheetah-random | $21.1 \pm 0.9$ | - | $\mathbf{34.5 \pm 1.1}$ |
| walker2d-random | $-0.3 \pm 0.1$ | - | $\mathbf{10.3 \pm 2.2}$ |
| hopper-random | $5.5 \pm 2.1$ | - | $\mathbf{31.7 \pm 0.9}$ |
| halfcheetah-medium | $47.6 \pm 0.3$ | $46 \pm 1$ | $\mathbf{62.1 \pm 0.5}$ |
| walker2d-medium | $86.3 \pm 0.3$ | $70 \pm 15$ | $\mathbf{97.2 \pm 2.5}$ |
| hopper-medium | $63.1 \pm 0.6$ | $65 \pm 10$ | $\mathbf{107.7 \pm 1.5}$ |
| halfcheetah-medium-replay | $46.1 \pm 0.3$ | $43 \pm 1$ | $\mathbf{56.8 \pm 1.2}$ |
| walker2d-medium-replay | $84.0 \pm 1.0$ | $46 \pm 19$ | $\mathbf{101.5 \pm 1.4}$ |
| hopper-medium-replay | $91.9 \pm 4.3$ | $53 \pm 9$ | $\mathbf{103.4 \pm 3.7}$ |
| halfcheetah-medium-expert | - | $75 \pm 16$ | $\mathbf{94.6 \pm 1.1}$ |
| walker2d-medium-expert | - | $110 \pm 0.5$ | $111.5 \pm 1.9$ |
| hopper-medium-expert | - | $103 \pm 14$ | $\mathbf{113.3 \pm 2.3}$ |

Table 9: The additional performance comparison between PGD (Jackson et al., 2024), DWM (Ding et al., 2024b) and ADEPT. The significant results are bolded.

The results are from their original paper cited by the reviewer. Note that PGD didn't report their performance on medium-expert dataset, while DWM didn't report their performance on random dataset and their results are rounded to the nearest whole number. Compared to these two results, our method shows significant advantages in every environment and dataset.

## C.7 HYPERPARAMETER SELECTION ON HORIZON

| env | $H = 1$ | $H = 5$ | $H = 20$ | $H = 50$ |
|---|---|---|---|---|
| halfcheetah-random | $19.5 \pm 0.2$ | $34.5 \pm 1.1$ | $37.1 \pm 0.5$ | $39.3 \pm 0.4$ |
| walker2d-random | $2.4 \pm 2.0$ | $10.3 \pm 2.2$ | $9.8 \pm 2.5$ | $5.9 \pm 3.9$ |
| hopper-random | $31.6 \pm 0.4$ | $31.7 \pm 0.9$ | $15.5 \pm 2.3$ | $9.8 \pm 5.4$ |
| halfcheetah-medium | $56.7 \pm 1.3$ | $62.1 \pm 0.5$ | $64.0 \pm 0.4$ | $67.7 \pm 1.6$ |
| walker2d-medium | $53.6 \pm 11.2$ | $97.2 \pm 2.5$ | $97.6 \pm 4.8$ | $94.4 \pm 3.9$ |
| hopper-medium | $0.1 \pm 0.1$ | $107.7 \pm 1.5$ | $103.6 \pm 1.0$ | $90.0 \pm 3.4$ |
| halfcheetah-medium-replay | $46.0 \pm 1.3$ | $56.8 \pm 1.2$ | $59.2 \pm 1.1$ | $67.7 \pm 1.6$ |
| walker2d-medium-replay | $1.4 \pm 1.1$ | $101.5 \pm 1.4$ | $96.3 \pm 2.5$ | $103.5 \pm 5.1$ |
| hopper-medium-replay | $101.8 \pm 0.5$ | $103.4 \pm 3.7$ | $91.4 \pm 5.2$ | $96.6 \pm 8.1$ |
| halfcheetah-medium-expert | $53.9 \pm 6.9$ | $94.6 \pm 1.1$ | $93.6 \pm 1.0$ | $90.0 \pm 3.4$ |
| walker2d-medium-expert | $0.4 \pm 0.2$ | $111.5 \pm 1.9$ | $100.9 \pm 3.0$ | $100.3 \pm 3.2$ |
| hopper-medium-expert | $0.1 \pm 0.1$ | $113.3 \pm 2.3$ | $97.8 \pm 5.6$ | $103.2 \pm 4.0$ |
| Average | $30.6 \pm 2.1$ | $77.1 \pm 1.7$ | $72.2 \pm 2.5$ | $72.4 \pm 3.7$ |

Table 10: Hyperparameter selection on Horizon $H$

Based on these results we found that the best pick of $H$ varies on different datasets and environments. For this study, the value of horizon is not the emphasis in our work, so we simply choose $H = 5$ as a reasonable value for all datasets.

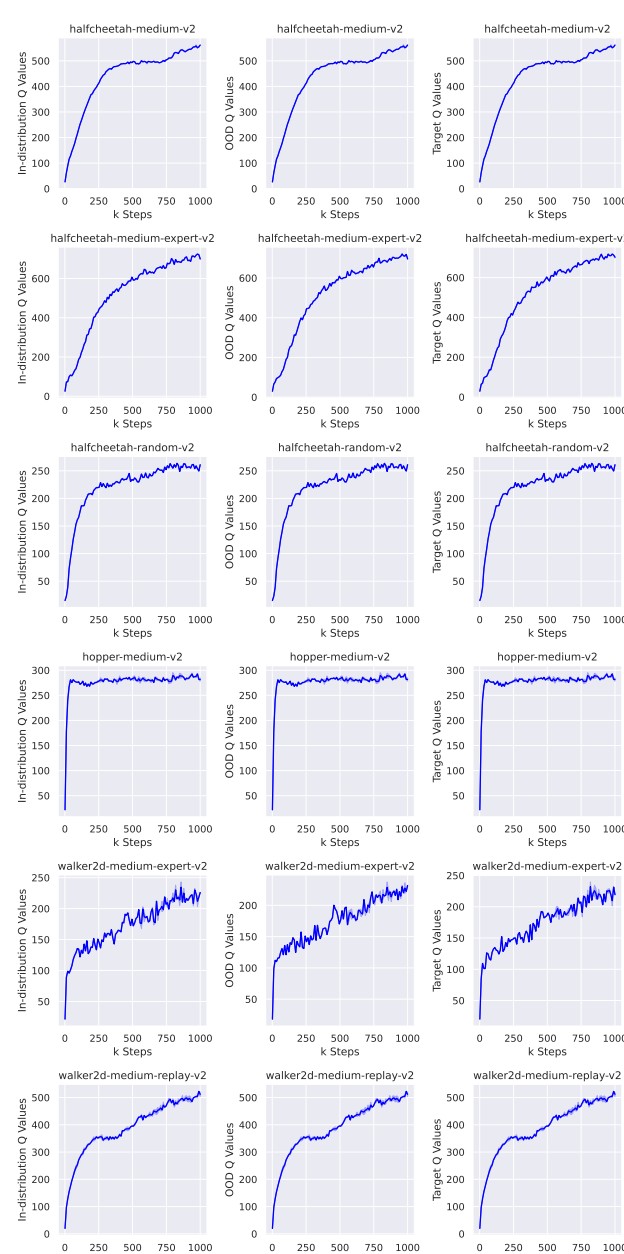

Figure 6: Q value estimation on in-distribution samples, out-of-distribution samples and target values.

## C.8    Q VALUES ON OUT-OF-DISTRIBUTION STATE AND ACTION PAIRS

To show that ADEPT help provide good and stable value estimation, we present additional results of several environments in Figure 6. Based on the traditional SAC architecture, we estimate the in-distribution Q-value by $\mathbb{E}_{(s,a)\sim\mathcal{D}}[\frac{Q_{\theta_1}+Q_{\theta_2}}{2}]$, the out-of-distribution Q-value by $\mathbb{E}_{s\sim\mathcal{D},a\sim\pi(\cdot|s)}[\frac{Q_{\theta_1}+Q_{\theta_2}}{2}]$, and denote the target Q value of the in-distribution samples. We find the Q estimation curve is stable and smooth in nearly all of the tests, showing that ADEPT largely mitigate the out-of-distribution overestimation issue.

