# OpenReview forum: "Offline Reinforcement Learning with Closed-loop Policy Evaluation and Diffusion World-Model Adaptation"
_ICLR.cc/2025/Conference — Submitted to ICLR 2025_

### Official Review · Reviewer_g96b · 2024-10-28

**Soundness:** 2
**Presentation:** 1
**Contribution:** 2
**Rating:** 5
**Confidence:** 2

**Summary:**

This paper deals with synthetic data generation in offline RL using diffusion models. Their principal contribution revolves around two methodological changes; introducing a penalty in the reward calculation for uncertain states, and employing importance sampling to account for policy distribution shift. They motivate their design decisions theoretically, and demonstrate strong performance in D4RL, a standard offline RL benchmark.

**Strengths:**

In general, I find the work reports good results and, while in places is hard to parse, has writing of a reasonable standard. I raise a number of strengths for this paper below.

- I find the related work has generally good coverage over crucial areas, besides what I would class as a significant error in description for [1] (below) and an omission.
- I like the approach of penalising overestimation by relying on the uncertainty of the world model itself. This is quite elegant.
- Using importance sampling to account for distribution shift is also intuitive (though based on the ablation has a relatively minimal impact on performance). I guess it just makes things theoretically sounder.
- I am grateful to see an ablation study, which I think is very inforamtive. However, I think making claims about the significance of important sampling is hard, given that reported values are often in confidence. The ablation study does clearly suggest that the uncertainty penalty in ADEPT contributes to performance improvement.

**Weaknesses:**

I have a number of strong concerns about this paper.

- I found figure 1 quite confusing, and personally think that if a figure needs such a long caption the figure is probably quite unclear. For instance, I don't follow what (b3) demonstrates. It seems like finding a way to show this without showing all the data points etc., for visual purposes, might make things clearer.
- In related work, stating that [1] doesn't 'provide explicit mechanisms to solve the distributional shift issue' is **fundamentally false** - that is the entire and explicit basis of the method. Besides this, I found the related work relatively thorough; one other relevant work would be [2].
- I found the description of the architecture hard to interpret. I would clarify that the MLP predicts the reward when the architecture is first interpreted. Similarly, the way the inputs are introduced ('we replace $\mathbf{x}_0$ with ...') was a bit confusing and could be worded better.
- Despite spending a long time with it, and attempting to verify the appendix proofs, I found I had a tough time with this maths and didn't find it intuitive to follow. It is also not made clear to me how the derivations in the paper lead to the reward gap guarantee at the top. Note I am not coming from a theoretical background, but imagine that others might also find this difficult.
- It feels this should be compared to other existing methods for compensating for overestimation bias. The key baselines here should be comparing the same policy optimisation algorithm with different approaches for reducing overcompensation bias. This is not what is shown in this paper.
- There are no error bars for any of the results besides ADEPT's, meaning it hard to see overlapping of confidence intervals.
- It is unclear whether the error bars report standard devision or standard derror. In table 2, the caption reads 'we show the standard deviation of the performance... Note that the standard *error* is usually large...'
- I feel it is important to raise the significant issue of referencing in this paper as a weakness as well as in my ethics review. Buried in the appendix (line 903) there are 12 cited papers, many with common authorship and none with relevance to this paper or explanation. Either these papers are relevant to this work, and should be raised as related work with explanation, or they are not and thus should not be included. I assume these papers should not be included in this paper, but if they should be describing **why** is important.


There are also a small number of minor points and typos to highlight:
- In line 37, stating that offline datasets typically consist of limited transitions is tautological; the offline dataset can't be infinite by nature.
- In line 44, the world model does not interact with the policy! The policy interacts with teh world model.
- The acronym of the algorithm (ADEPT) does not fit its name at all really.
- Defining in line 181 that $\hat{P}$ is the transition distribution defined by the world model would be worthwhile.
- Line 190 'is a certain various schedule' doesn't make sense and I am not sure what it is meant to say.
- Line 190 'the diffusion model define another' - firstly, this should be 'defines another'. Secondly, the diffusion model is composed of a forward and backward process, rather than this being defined by the diffusion model itself.
- Line 199 does not make sense.
- The first sentence of the methodology does not make sense.
- Line 346: I don't really know what this means - what is the discrepancy?
- Line 357: 'of between $\pi$' should not have 'of'
- Line 381: 'there existing $\delta$' should read 'there exists $\delta$
- Line 398: 'piratically' is not the correct word I asume.
- Line 411: I don't know what $H$ is and it is not defined I don't think.
- In table 2, all bolded values are with a standard error of each so bolding, which implies significant improvement, is misleading.


[1] Policy Guided Diffusion, Jackson et al. 2024
[2] World Models via Policy-Guided Trajectory Diffusion, Rigter et al. 2024

**Questions:**

- It seems a bit counterintuitive/non-obvious to compute your done flag as when uncertainty reaches a limit. Does this effectively just lead to truncated rollouts? What kind of bias does this have on bias?
- I don't quite understand the assumption of line 340, that you can omit one of the inputs in the proof. To my understanding the reward model comes after the denoising model, and as such would not intrinsically be able to ignore $\hat{s}_{t+1}$. Is this not correct?
- How were hyperparameters for the underlying learning method tuned? It says they were consistent for different methods - were they tuned for ADEPT or for one of the base algorithms, or are they taking default values.
- How does this method compare to other approaches which compensate for overestimation bias? For example, how does it compare against policy guided diffusion ([1] above)?

**Details Of Ethics Concerns:**

I am concerned about the referencing used by the authors in this paper. In the appendix (B.1 Baselines) without any detail, there is a long list of cited works (12 papers) which seem to hold no relevance to the paper and all have some degree of common authorship. These are not mentioned elsewhere in the paper and are cited without explanation. This is done in line 903 of the paper:

> In addition, we cite to the following works. (Zhang et al., 2024b; Zou et al., 2024; Gao et al., 2024; Zhang et al., 2024a; Wáng et al., 2023; Fang et al., 2022; 2023; Zhou et al., 2022; 2023; Mei et al., 2023; Chen et al., 2023; 2021)

Observing some of these papers, it is clear that they have no relevance to this paper. For example, one of the cited papers is (line 679):

> Yì Xiáng J Wáng, Zhi-Hui Lu, Jason CS Leung, Ze-Yu Fang, and Timothy CY Kwok. Osteoporotic-
like vertebral fracture with less than 20% height loss is associated with increased further vertebral
fracture risk in older women: the mros and msos (hong kong) year-18 follow-up radiograph results.
Quantitative Imaging in Medicine and Surgery, 13(2):1115, 2023.

Besides the above, a large amount of the papers are about reinforcement learning but in completely different areas to this work (such as MARL). It would be one thing if these papers had been motivated with explanation, and as such had some link to the research at hand. However, their means of introduction and the way they are buried in a subsection of the appendix makes me believe that this is a case of academic dishonesty; in particular, this is an example of self-citation to boost the author's citation count rather than being relevant to the paper, and also risks affecting the paper's anonymity.

---

> ### Author Response · Authors · 2024-11-24
> **Author Response 1**
>
> We thank the reviewer for their thoughtful comments. We address your concerns below.
>
> **Weakness**:
>
> >I found figure 1 quite confusing, and personally think that if a figure needs such a long caption the figure is probably quite unclear. For instance, I don't follow what (b3) demonstrates. It seems like finding a way to show this without showing all the data points etc., for visual purposes, might make things clearer.
>
> Long captions are often used to make a figure self-contained. It is a common practice and, in our opinion, shouldn’t be considered as a weakness. We decided to show the data points in Figure 1 because the diffusion model is trained on the samples reweighted by importance sampling. Further explanations can be found in our main text. Nevertheless, we have adjusted Figure 1 in the revision, also based on the opinions of Reviewer PaFJ. We hope that the new figure will be easier to understand.
>
> >In related work, stating that [1] doesn't 'provide explicit mechanisms to solve the distributional shift issue' is fundamentally false - that is the entire and explicit basis of the method. Besides this, I found the related work relatively thorough; one other relevant work would be [2].
>
> The distribution shift problem considered in our paper is quite specific. As training goes on, not only is the target policy being updated and continuing to deviate from the behavior policy for data collection, the diffusion model – which is trained on offline data collected by behavior policy – also continues to deviate and leads to increased error in evaluating the target policy. Paper [1] did not consider this problem. In fact, taking a closer look at Paper [1], you can see that it didn’t show significant improvement compared with the unguided diffusion generated datasets or original datasets in nearly 2/3 of the environments, nor did it show higher performance than the previous work SyntheR [2] on the same dataset. The second paper you mentioned considers online RL, whereas we consider offline RL in this paper. We are happy to clarify this in the updated version to avoid such misunderstanding.
>
> > I found the description of the architecture hard to interpret. I would clarify that the MLP predicts the reward when the architecture is first interpreted. Similarly, the way the inputs are introduced ('we replace $x_0$ with ...') was a bit confusing and could be worded better.
>
> We feel the fact that the MLP generates the reward prediction is quite straightforward (so is the the inputs of the NNs), but can add a sentence on it as requested by the reviewer.
>
> > Despite spending a long time with it, and attempting to verify the appendix proofs, I found I had a tough time with this maths and didn't find it intuitive to follow. It is also not made clear to me how the derivations in the paper lead to the reward gap guarantee at the top. Note I am not coming from a theoretical background, but imagine that others might also find this difficult.
>
> Sorry to hear that the reviewer was not able to go through the entire proof in the appendix. As the reviewer mentioned, it may not be easy for someone who is not used to this type of analysis in this area. We’d suggest the reviewer start with the proofs in MOPO[1], MOReL[2], and [3], which have considered some partial special cases of our problem. Due to space limitations, we couldn’t add too much explanation to the paper. However, if the reviewer has specific questions about any steps of our proof, we will be happy to provide further information.
>
> >It feels this should be compared to other existing methods for compensating for overestimation bias. The key baselines here should be comparing the same policy optimisation algorithm with different approaches for reducing overcompensation bias. This is not what is shown in this paper.
>
> We strongly disagree with this comment. First of all, we are not sure where the terms “overestimation bias” or “overcompensation bias” are coming from. These are not related to our paper. As we introduced in the paper, the distributional shift problem is the fundamental reason for overestimation of returns, which is well known for offline reinforcement learning. We propose a new offline RL algorithm to address this problem. All baselines we considered in the evaluation, including model-free and model-based offline RL algorithms, are designed with different approaches to address this problem. Therefore, the claim that “This is not what is shown in this paper” is totally false.
>
> > There are no error bars for any of the results besides ADEPT's, meaning it hard to see overlapping of confidence intervals.
>
> We explain the reason for no error bars in the original paper in the response to Reviewer TBhS and Reviewer C8UH. As it's required by other reviewers, we have included variances of the baselines in the revision.

---

> ### Author Response · Authors · 2024-11-24
> **Author Response 2**
>
> >It is unclear whether the error bars report standard devision or standard derror. In table 2, the caption reads 'we show the standard deviation of the performance... Note that the standard error is usually large...'
>
> It is clear that we report the standard deviation in our results like other relevant works in this area. We believe that there’s no issue in this caption since the high standard error is exactly the reason that causes the high standard deviation of our results.
>
> >I feel it is important to raise the significant issue of referencing in this paper as a weakness as well as in my ethics review. Buried in the appendix (line 903) there are 12 cited papers, many with common authorship and none with relevance to this paper or explanation. Either these papers are relevant to this work, and should be raised as related work with explanation, or they are not and thus should not be included. I assume these papers should not be included in this paper, but if they should be describing why is important.
>
> This is a mistake. As the reviewer noted, some of these papers are on completely different topics like bio-data processing. These were legacies from the previous template/paper we used. We tried to remove them – successfully from the main text – but seem to have missed one place in the appendix. It's been removed in the revision and there's no need to read too much into it.
>
> **Minors**:
>
> Thank you so much for reading our work thoroughly and catching these. Most of the minors have been corrected in the revision. However, we disagree with the following ones:
>
> >In line 44, the world model does not interact with the policy! The policy interacts with teh world model.
>
> This claim doesn't make any sense, since the word “interaction” naturally implies a two-way relationship.
>
> >The acronym of the algorithm (ADEPT) does not fit its name at all really.
>
> That's not an issue, since the name of an algorithm is not necessarily its acronym.
>
> > The first sentence of the methodology does not make sense.
>
> We didn't see any problem in that sentence. Could you make it more specific?
>
> > Line 346: I don't really know what this means - what is the discrepancy?
>
> Discrepancy is already defined in section 4.1 text without an equation index.
>
> >Line 411: I don't know what $H$ is and it is not defined I don't think.
>
> $H$ is already defined in Section 3 as the horizon, Paragraph named “Offline RL using World Models”.
>
> >In table 2, all bolded values are with a standard error of each so bolding, which implies significant improvement, is misleading.
>
> This sentence seems somehow confusing. Nevertheless, we revised our paper to bold only the significant results.
>
> **Questions**:
>
> >It seems a bit counterintuitive/non-obvious to compute your done flag when uncertainty reaches a limit. Does this effectively just lead to truncated rollouts? What kind of bias does this have on bias?
>
> When the estimated uncertainty is larger than a threshold, a done flag will be set and truncate the rollout. A potential issue of this mechanism is that the truncated rollouts not only prevent synthetic trajectories from moving to uncertain states and action pairs on which the world model might have high prediction errors but also add a penalty in reward for those pairs, leading the target policy to be more conservative. Previous methods that sample initial states from any time step in the offline dataset could have another problem that the synthetic trajectories can’t move too far away from the existing trajectories. In our method, we sample initial states from both the offline dataset and the replay buffer consisting of previous synthetic trajectories to solve this problem.
>
> > I don't quite understand the assumption of line 340, that you can omit one of the inputs in the proof. To my understanding, the reward model comes after the denoising model, and as such would not intrinsically be able to ignore $\hat{s} _{t+1}$. Is this not correct?
>
> For our method, it could be more accurate if we use $r_{\eta, \theta}(s_t, a_t) = \sum_{s’} r_{\eta}(s_t, a_t | s’)P_\theta(s’|s_t, a_t)$ to represent the reward prediction model in the theory, by treating the diffusion process and MLP as a whole. However, we simplified that as $r_{\eta}(s_t, a_t)$ for more conciseness and generalization, while such a simplification will not have any impact on the proof.
>
> > How were hyperparameters for the underlying learning method tuned? It says they were consistent for different methods - were they tuned for ADEPT or for one of the base algorithms, or are they taking default values.
>
> We use the default hyperparameters for policy optimization (SAC) and most of the diffusion model training. We tuned the hyperparameters for uncertainty penalty($\lambda_r$, $\lambda_0$), the denoising steps ($K$), and the width of the diffusion network. The results can be found in the appendix.

---

> ### Author Response · Authors · 2024-11-24
> **Author Response 3**
>
> >How does this method compare to other approaches which compensate for overestimation bias? For example, how does it compare against policy guided diffusion ([1] above)?
>
> We attached the comparison results in the response to Reviewer C8UH. Please check that.
>
> [1] Policy Guided Diffusion, Jackson et al. 2024
> [2] Lu, Cong, et al. "Synthetic experience replay." Advances in Neural Information Processing Systems 36 (2024).
> [3] Janner, Michael, et al. "When to trust your model: Model-based policy optimization." Advances in neural information processing systems 32 (2019).

---

> ### Comment · Reviewer_g96b · 2024-11-25
>
> Dear authors,
>
> Thank you for your very thorough response. Below, I go through step-by-step:
>
> > we have adjusted Figure 1 in the revision
>
> Given it is at the beginning of the paper, it is not helpful for understanding the frequency of things changing - for instance, why $\hat{\mathcal{M}}$ changes only once. I personally do not think it is as helpful for conveying the idea of the paper as a simpler figure would be. I do however look forward to seeing the new version.
>
> > Paper [1] did not consider this problem
>
> To my understanding, this is the exact problem considered by [1]. While I appreciate that your results may be better (in fact, they are - and I appreciate the additional comparison), that is not an excuse for falsehoods in related work comparison.
>
> > We feel the fact that the MLP generates the reward prediction is quite straightforward
>
> Perhaps this could be clearer by stating 'and reward function $R(s_t,a_t)$ **respectively**'?
>
> > However, if the reviewer has specific questions about any steps of our proof, we will be happy to provide further information.
>
> Note that I have not counted anything in the proof within my review. I will spend some time in the next 24 hours to spend longer checking the maths to ensure my understanding is correct.
>
> > First of all, we are not sure where the terms “overestimation bias”
>
> I apologise for my missing reference, the term overestimation bias comes from: "The Edge-of-Reach Problem in Offline Model-Based Reinforcement Learning", Sims et al.
>
> > Therefore, the claim that “This is not what is shown in this paper” is totally false.
>
> I apologise for any lack of clarity; by this, I mean that since SAC is the policy optimisation algorithm in this work it feels like there should be additional comparison of SAC with other methods for preventing overestimation of values, such as adding pessimism to the value (as in the paper I referenced above).
>
> > As it's required by other reviewers, we have included variances of the baselines in the revision.
>
> Thank you for this. I will wait until the new version of the pdf is uploaded before making any edits.
>
> > We believe that there’s no issue in this caption since the high standard error is exactly the reason that causes the high standard deviation of our results.
>
> The confusion arose for table 2 as the caption discusses both standard deviation and error. For clarity's sake, I would refer consistently to standard deviation but will not hold this against the paper.
>
> > It's been removed in the revision and there's no need to read too much into it.
>
> I am glad to know that this was a mistake and has been corrected.
>
> > since the word “interaction” naturally implies a two-way relationship.
>
> Sure, but conventionally the 'agent interacts with the environment' makes more sense due to the suggestion of an active subject and passive object. While it is true that the environment in turn interacts wiht the agent, it feels clunky to write this was as the agent is the active one in the relationship.
>
> > That's not an issue, since the name of an algorithm is not necessarily its acronym.
>
> I only flagged this for stylistic reasons and do not consider it in my score (and agree that it would be wrong to as it is not necessary for the anem and acronym to match).
>
> > We didn't see any problem in that sentence.
>
> Apologies. I think my confusion arises due to the : introducing a list, but then the sentence continuing after; due to the colon, it reads as three things in a list, and doesn't amke grammatical sense to continue the sentence after the second item in the list.  I think: ' As shown in Figure 2, the two key components in ADEPT are policy evaluation on a guided diffusion world model and importance-sampled world model updates. These work in a closed-loop operation through the whole training process' would read better personally, for instance.
>
> > Discrepancy is already defined in section 4.1
>
> Thank you for clarifying.
>
> > already defined in Section 3 as the horizon, Paragraph named “Offline RL using World Models”.
>
> I apologise for missing this, $H$ is correctly defined and I missed this on my read through.
>
> > This sentence seems somehow confusing. Nevertheless, we revised our paper to bold only the significant results.
>
> I apologise for any confusion caused, but appreciate your new bolding of signifcance and look forward to seeing the new manuscript.
>
> > (Response to all questions)
>
> These are enlightening and fit what I was expecting. THank you for your clarification.
>
> > We attached the comparison results in the response to Reviewer C8UH. Please check that.
>
> Thank you. It certainly seems like ADEPT has significant improvement over PGD.

---

> > ### Comment · Reviewer_g96b · 2024-11-25
> >
> > Overall, I believe the authors have made significant effors during the discussion period to engage in the criticism presented. While I will not make any adjustments to my score until seeing the amended manuscript, I am leaning towards increasing my score from 3 to either 5 or 6 for this paper to reflect the new results and clearer writing. I will make this judgement upon seeing the new paper.

---

> > ### Author Response · Authors · 2024-11-25
> >
> > Thank you. We truly appreciate the quick response! Please see our further clarification below:
> >
> > > Re: Figure 1.
> >
> > Yes, we hear you. We can move Figure 1 to a later place in the paper, perhaps as an illustration when we describe the algorithm.
> >
> > > Re: Reference [1].
> >
> > We strongly believe that paper [1] is not solving the problem we described in the response. In particular, the diffusion model – which is trained on offline data collected by behavior policy – also continues to deviate and leads to increased error in evaluating the target policy. This was not considered in [1] and is the main reason for us to achieve much improved performance, as you correctly pointed out. But if the reviewer finds our original wording in the paper confusing. We will be happy to revise it and clarify.
> >
> > >Re: "Note that I have not counted anything in the proof within my review. I will spend some time in the next 24 hours to spend longer checking the maths to ensure my understanding is correct."
> >
> > Sure. If there is any additional info we can provide, please let us know. In the original submission, we have divided the proof into multiple lemmas and theorems, to help improve readability. Please feel free to check the proof flow/logic first, before delving into the details of each lemma/theorem. It may help.
> >
> > >Re: Reward function
> >
> > Yes, we add this clarification in the paper.
> >
> > > Re: “overestimation bias”
> >
> > The paper you mentioned is addressing an overestimation bias problem by adding pessimism. While we use SAC, the problem we are addressing in this paper is indeed a different one. The distribution shift problem is due to the fact that as training goes on, not only is the target policy being updated and continuing to deviate from the behavior policy for data collection, the diffusion model – which is trained on offline data collected by behavior policy – also continues to deviate and leads to increased error in evaluating the target policy. Thus we propose an adaptive algorithm to update the diffusion model jointly during policy evaluation, thus producing better estimates of the policy reward and updates. We will further clarify it in the paper.
> >
> > In summary, we believe this is a strong paper solving a very important problem in joint policy evaluation and diffusion model adaptation. We certainly hope to clarify any remaining misunderstanding and share our results with the community.
> >
> > Thanks again!

---

> > > ### Comment · Reviewer_g96b · 2024-11-26
> > >
> > > Dear Authors,
> > >
> > > Thank you for your prompt response.
> > >
> > > > This was not considered in [1]
> > >
> > > Like I say, this is my understanding of exactly what [1] focuses on - it incorporates a guidance term to the diffusion model such that samples generated are more 'on-policy' for the updated policy.
> > >
> > > > the problem we are addressing in this paper is indeed a different one
> > >
> > > Right, thank you for your clarification.
> > >
> > > I look forward to seeing the new version of the paper, which I will base my updated assessment on. Please do drop a comment when it is out so I can check the leftover points before considering whether to increase my score.

---

> > > > ### Comment · Reviewer_g96b · 2024-12-02
> > > >
> > > > Dear authors,
> > > >
> > > > I apologise for my delay in processing the new manuscript. I have now spent some time going through this - albeit not enough to fully process it - but wanted to get a response in before the deadline. I will continue to go through edits before discussing with the AC and other reviewers, so do not fear if I have missed anything.
> > > >
> > > > I am happy to see additional baselines included, as well as error bars for the results. Out of interest, in table 1 I do not think ADEPT can claim statistical significance on halfcheetah med-exp due to overlapping bounds with many other methods. This does beg the question - is there anything about the distribution in halfcheetah which leads to ADEPT underperforming quite consistently. While I appreciate this is a limitation of D4RL MuJoCo, it would be good to see this tested over additional *environments*, rather than just dataset characteristics, to see if halfcheetah is an anomaly or more typical.
> > > >
> > > > While Figure 1 now has a lot going on, I believe it adds additional clarity over what was originally included - and the slight changes to (now) Figure 2 also help.
> > > >
> > > > As I say, I will continue to look at the smaller changes introduced to the paper in the coming days. As it stands, I am happy to increase my score from 3 to 5 - I still hold some reservations, which I have pointed out above regarding the way [1] is discussed and in what I perceive to be a slight lack of baselines, but I feel that there have been significant improvements which warrant a score higher than rejection. However, I am also going to lower the confidence of my review, since there are still some parts of the updated paper which I have not yet had the time to read (due to the fact it was uploaded close to the deadline). While I don't think I can post after today, I will bear any extra information in mind during the next stage of the peer review (and will update any scores accordingly, if I can and believe it justified).

---

### Official Review · Reviewer_C8UH · 2024-11-01

**Soundness:** 3
**Presentation:** 2
**Contribution:** 3
**Rating:** 6
**Confidence:** 4

**Summary:**

This paper proposes a model-based offline RL algorithm leveraging diffusion models. Unlike past works that use pre-trained, frozen diffusion models for generating synthetic rollouts, this work proposes to iteratively align the model's predictions with the evolving policy during training. A theoretical analysis is provided that provides an upper bound on the return gap between an optimal policy trained inside the diffusion model versus the real environment, and experiments are performed on D4RL that show an improvement over canonical offline RL methods.

**Strengths:**

- The experimental results are strong. Most of the required baselines (see weaknesses) are implemented and the proposed method outperforms them in aggregate.
- The theoretical analysis is additive, and as far as I can tell, sound.
- The paper is generally well-written and well-motivated.
- The appendix provides some interesting additional results that support the main body.

**Weaknesses:**

Though the results are impressive, my primary concern is that the method appears to be very similar to that proposed in [1,2]. It seems the key difference from prior works is the proposed mechanism for realigning the diffusion model's predictions with the changing policy during training. However, when this mechanism is ablated in Figure 3, it appears to only significantly improve performance in one of the four datasets on halfcheetah (medium-replay). The reader would be able to better understand the performance gains expected from this method w.r.t. the methods from [1,2] if they were implemented as baselines, but unfortunately they are not. A more thorough comparison of the authors proposed method with those from [1,2] would improve the paper, and leave the reader more confident that their proposals are a concrete step forward from these works.

**Minor feedback**

- Lines 120-121: Ha & Schmidhuber's paper introduced world models for online RL, not offline RL
- Section 3 title should be "Preliminaries"
- Line 151: "Agent [that] acts..."
- Line 153: transits -> transitions
- Line 161: "real dynamics $P$"?
- Line 272: collects -> collected
- Section 5 title should be "Experiments"
- Line 490 there should be text following _i.e._
- Missing closed brackets in Equations 7 and 8
- Missing brackets in Equation 12
- Line 398: "practically"

**References**

[1] Zihan Ding, Amy Zhang, Yuandong Tian, and Qinqing Zheng. Diffusion world model. arXiv preprint
arXiv:2402.03570, 2024.

[2] Matthew Thomas Jackson, Michael Tryfan Matthews, Cong Lu, Benjamin Ellis, Shimon Whiteson,
and Jakob Foerster. Policy-guided diffusion. arXiv preprint arXiv:2404.06356, 2024

**Questions:**

**Important questions**

- How does the performance of ADEPT compare to past works that utilise diffusion models as world models for policy training on D4RL (namely DWM (Ding et al., 2024) or PGD (Jackson et al., 2024)?
- Are the importance-sample updates more important to final performance in environments other than halfcheetah?

**Less important questions**

- Where are the uncertainty intervals for the baselines in Tables 1 and 2?
- You set $H$ to 5, how does performance change with different values of $H$?

If the authors answer the important questions I'm more than happy to update my score.

---

> ### Author Response · Authors · 2024-11-24
> **Author Response 1**
>
> We thank you for evaluating our work. We now answer the following concerns raised in the review.
>
> **Minor feedback**:
> Thanks for catching these! We have addressed these in the new revision.
>
> **Weakness**:
> > Though the results are impressive, my primary concern is that the method appears to be very similar to that proposed in [1,2]. It seems the key difference from prior works is the proposed mechanism for realigning the diffusion model's predictions with the changing policy during training. However, when this mechanism is ablated in Figure 3, it appears to only significantly improve performance in one of the four datasets on halfcheetah (medium-replay). The reader would be able to better understand the performance gains expected from this method w.r.t. the methods from [1,2] if they were implemented as baselines, but unfortunately they are not. A more thorough comparison of the authors proposed method with those from [1,2] would improve the paper, and leave the reader more confident that their proposals are a concrete step forward from these works.
>
> Although ADEPT, DWM and PGD both use diffusion models as the world model in offline reinforcement learning, there are still many fundamental differences. The uniqueness of ADEPT comes from 2 key components as we mentioned in the paper: the uncertainty estimation and penalty based on the diffusion model itself, and the world model alignment via importance sampling. To the best of our knowledge, we are the first to apply these two methods in model-based offline RL and provide theoretical analysis for that. Besides, DWM and PGD design their diffusion model as multi-step planners, while we adopt it as a single-step planner for policy evaluation.
>
> **Questions**:
> >How does the performance of ADEPT compare to past works that utilise diffusion models as world models for policy training on D4RL (namely DWM (Ding et al., 2024) or PGD (Jackson et al., 2024)?
>
> The performance comparison of ADEPT and two works mentioned are as follows:
> | env | PGD | DWM | ADEPT |
> |-----------|-----------|-----------|-----------|
> | halfcheetah-random | $21.1\pm0.9$ | - | $\mathbf{34.5\pm1.1}$ |
> | walker2d-random | $-0.3\pm0.1$ | - | $\mathbf{10.3\pm2.2}$ |
> | hopper-random | $5.5\pm2.1$ | - | $\mathbf{31.7\pm0.9}$ |
> | halfcheetah-medium | $47.6\pm0.3$ | $46\pm1$ | $\mathbf{62.1\pm0.5}$ |
> | walker2d-medium | $86.3\pm0.3$ | $70\pm15$ | $\mathbf{97.2\pm2.5}$ |
> | hopper-medium | $63.1\pm0.6$ | $65\pm10$ |$\mathbf{107.7\pm1.5}$ |
> | halfcheetah-medium-replay | $46.1\pm0.3$ | $43\pm1$ | $\mathbf{56.8\pm1.2}$ |
> | walker2d-medium-replay | $84.0\pm1.0$ | $46\pm19$ | $\mathbf{101.5\pm1.4}$ |
> | hopper-medium-replay | $91.9\pm4.3$ | $53\pm9$ | $\mathbf{103.4\pm3.7}$
> | halfcheetah-medium-expert | - | $75\pm16$ | $\mathbf{94.6\pm1.1}$ |
> | walker2d-medium-expert | - | $110\pm0.5$ | $111.5\pm1.9$ |
> | hopper-medium-expert | - | $103\pm14$ | $\mathbf{113.3\pm2.3}$ |
>
> The results are from their original paper cited by the reviewer. Note that PGD didn’t report their performance on medium-expert dataset, while DWM didn’t report their performance on random dataset and their results are rounded to the nearest whole number. Compared to these two results, our method shows significant advantages in every environment and dataset.
>
> >Are the importance-sample updates more important to final performance in environments other than halfcheetah?
>
> We provided additional results in the revision presented with aggregated statistics as suggested by Reviewer PaFJ. We observed a similar phenomenon in the other two environments that importance sampling mainly improves the performance on medium-replay dataset, where the behavior policy is more stochastic and the policy shifting problem is more severe ($\hat{\varepsilon}_p (\pi)$ is high), while having limited effect on datasets collected by more determined and nearly optimal policies as medium-expert ($\hat{\varepsilon}_p (\pi)$ is low). These results further support our theory that the importance sampling method could largely solve the distributional shift problem by continuously aligning the world model with the new distribution under the target policy.
>
> >Where are the uncertainty intervals for the baselines in Tables 1 and 2?
>
> Thanks for highlighting that. The intervals are not included at first because we follow the presentation style of existing works on diffusion-based offline RL [1-3] and try to make the table look more concise. In the revision, we add the uncertainty intervals for baselines in Table 1. However, the original papers on baselines didn’t provide the uncertainty intervals for environments in Table 2. Thus, we decide to move it to the Appendix as secondary results.

---

> ### Author Response · Authors · 2024-11-24
> **Author Response 2**
>
> >You set $H$ to 5, how does performance change with different values of $H$?
>
> That’s a very good point. The value of $H$ could affect the distribution of synthetic policy significantly since it generally determines how far the trajectory could deviate from the offline dataset. Setting a large H has its pros and cons. On the one hand, the model-generated data could be closer to the true distribution under the target policy. However, compounding errors will grow rapidly even with uncertainty estimation, which could counteract the benefit and get exploited by the policy to gain plausibly high returns.
> We select different values of H and show the results as follows:
>
> | env | $H=1$ | $H=5$ | $H=20$ | $H=50$ |
> |-----------|-----------|-----------|-----------|-----------|
> |halfcheetah-random|$19.5\pm0.2$|$34.5\pm1.1$|$37.1\pm0.5$|$39.3\pm0.4$|
> |walker2d-random|$2.4\pm2.0$|$10.3\pm2.2$|$9.8\pm2.5$|$5.9\pm3.9$|
> |hopper-random|$31.6\pm0.4$|$31.7\pm0.9$|$15.5\pm2.3$|$9.8\pm5.4$|
> |halfcheetah-medium|$56.7\pm1.3$|$62.1\pm0.5$|$64.0\pm0.4$|$67.7\pm1.6$|
> |walker2d-medium|$53.6\pm11.2$|$97.2\pm2.5$|$97.6\pm4.8$|$94.4\pm3.9$|
> |hopper-medium|$0.1\pm0.1$|$107.7\pm1.5$|$103.6\pm1.0$|$90.0\pm3.4$|
> |halfcheetah-medium-replay|$46.0\pm1.3$|$56.8\pm1.2$|$59.2\pm1.1$|$67.7\pm1.6$|
> |walker2d-medium-replay|$1.4\pm1.1$|$101.5\pm1.4$|$96.3\pm2.5$|$103.5\pm5.1$|
> |hopper-medium-replay|$101.8\pm0.5$|$103.4\pm3.7$|$91.4\pm5.2$|$96.6\pm8.1$|
> |halfcheetah-medium-expert|$53.9\pm6.9$|$94.6\pm1.1$|$93.6\pm1.0$|$90.0\pm3.4$|
> |walker2d-medium-expert|$0.4\pm0.2$|$111.5\pm1.9$|$100.9\pm3.0$|$100.3\pm3.2$|
> |hopper-medium-expert|$0.1\pm0.1$|$113.3\pm2.3$|$97.8\pm5.6$|$103.2\pm4.0$|
> |Average|$30.6\pm2.1$|$77.1\pm1.7$|$72.2\pm2.5$|$72.4\pm3.7$|
>
> Based on these results we found that the best pick of $H$ varies on different datasets and environments. For this study, the value of horizon is not the emphasis in our work, so we simply choose $H=5$ as a reasonable value for all datasets.
>
>
> [1] Janner, Michael, et al. "Planning with Diffusion for Flexible Behavior Synthesis." International Conference on Machine Learning. PMLR, 2022.
>
> [2] Ajay, Anurag, et al. "Is Conditional Generative Modeling all you need for Decision Making?." The 11th International Conference on Learning Representations (ICLR), 2023.
>
> [3] Wang, Zhendong, Jonathan J. Hunt, and Mingyuan Zhou. "Diffusion Policies as an Expressive Policy Class for Offline Reinforcement Learning." The 11th International Conference on Learning Representations (ICLR) 2023.

---

> ### Comment · Reviewer_C8UH · 2024-11-29
> **Thanks for your updates**
>
> Hi authors, thanks very much for your hard work addressing my (and the other reviewers') questions.
>
> You have responded convincingly to my important questions which I appreciate. In particular, the extended ablation study (Section 5.2 / Figure 4) is a significant improvement on the original paper. I notice that you have not included the above comparison to PGD and DWM in the updated paper; I would recommend that you add this, even if just as an appendix, because these are important recent works that readers will be keen to see a comparison against.
>
> Because you have addressed my important questions I will update my score from 5 to 6 accordingly.
>
> Finally, I was concerned by Reviewer g96b's findings that you had cited a series of irrelevant papers in the appendix in what looked like an attempt to boost citations. Clearly this behaviour can't help but raise doubts about the scientific integrity of the rest of the paper. I appreciate you have stated this was a mistake, but it not clear how such citations could appear without the authors' knowledge. I'm not sure what the protocol is here, so I will leave it to the AC to decide how to proceed.

---

> > ### Author Response · Authors · 2024-11-29
> >
> > Thank you very much for your helpful feedback and for increasing your score!
> >
> > The comparison to PGD and DWM has already been added in Appendix C.6, Table 9 of the revision. Also, as we stated in the response to Reviewer g96b, these citations were legacies from the previous template we used. We do know the consequences of this mistake, and we apologize for omitting this issue in the appendix. It has been deleted completely in the revision.

---

> > > ### Comment · Reviewer_C8UH · 2024-11-29
> > >
> > > Thanks for your reply, and apologies for missing that you had already included these new results in the appendix--my mistake!

---

### Official Review · Reviewer_TBhS · 2024-11-03

**Soundness:** 2
**Presentation:** 2
**Contribution:** 3
**Rating:** 3
**Confidence:** 3

**Summary:**

This paper operates in the setting of offline reinforcement learning. The paper proposes a new approach that uses an uncertainty-penalized diffusion model as a world model. This world model is used to update the offline policy by constraining a standard SAC's actions via uncertainty estimates. The world model is updated using importance sampling to address the distribution shift of the trained policy from the behavioral policy over time. The paper provides a theoretical analysis of error bounds as well as an experimental section highlighting the approaches performance in comparison with recent offline methods.

**Strengths:**

I would like to preface this review by saying that I am not an expert in offline model-based RL. However, I am very familiar with the general online RL and theoretical RL landscape.

1. Clarity
a) The language in the paper is largely clear and the text has a clear red line that the reader can follow.
b) The visualizations in Figure 1 and 2 are helpful to understand the approach.

2. Related work
a) From what I can tell, the work cites the most prominent approaches in offline (model-based) RL and provides a reasonable amount of related work to differentiate its contribution from prior art.

3. Novelty
a) Based on my (incomplete) knowledge the idea to constrain a model based on the distribution shift of the policy based on offline data only seems novel enough to warrant publication. However, other reviewers may have more insight into this than I do.

4. Experiments
a) The experiments are conducted with a sizable number of baselines to demonstrate the capabilities. I do think the experiments demonstrate that there might be benefits of the method in lower quality data regimes. However, I have several things that need to be addressed before I can make a certain statement about this. I will come to them later.

**Weaknesses:**

1. Mathematical rigor and theory
a) It is a well-known fact in the learning theory literature that approximate models are sufficient to obtain high return. Analysis on this goes as far back as [1] which can easily be adjusted to fixed distributions. Given that the paper states that this analysis is based on prior work, it is unclear to me what is being claimed to be novel here. There might be subtleties I am missing due to unclarity of notation which I will outline next.
b) In equation 5, and all following similar expectations, it is not clear what $s_t$ is sampled from. This is quite important given that we are talking about distribution shifts and without this notation being precise it is difficult to determine the correctness.  It is also not clear to me what an expectation over $\mathcal{M}$ means which seems to be a set.
c) In equation 6, the TV distance is ill defined since $R(s_t, a_t)$ is not a distribution and there seem to be no assumptions on this function anywhere else.
d) It is unclear to me, why assumption 4.3 is reasonable. I will ask for clarification.
e) Theorem statement should generally be concise, but they should also be self-contained. In order to understand Theorem 4.5, one would have to read large parts of the paper just to understand the notation. I recommend adjusting this as needed for readability.  The provided proof is also not a proof, but it looks more like a sketch. I recommend stating it as a sketch and referring to the full proof.

2. Experiments
a) Experiments over 5 seeds in reinforcement learning can often be misleading given the high variance.
b) Tables 1 and 2 have the maximum average bolded. This can be misleading as the reader might think these methods are superior as it is not uncommon to bold statistically significant results rather than max averages. I recommend the manuscript is switched to the latter to avoid confusion.
c) To address the previous point, it is necessary to report variance measures for all baselines and not the presented algorithm. That should in general always be the case. In Table 1, all favorable results on the med-exp dataset are within variance of another approach, at least one of the favorable results of med-rep is within variance of another approach and it is unclear how many of the other results are significant. In Table 2, at least 5/6 results seem to be within variance.  Thus, the claim that the provided algorithm outperforms existing SOTA algorithms is not well supported.
d) The paper does not provide any additional analysis besides best returns on D4RL and as a result, it is not clear when I should use this method as the results on lower quality datasets are not completely consistent. This makes things tricky because many of the other results may not be significant. One way to remedy the fact that the results are not necessarily much stronger in many cases would be to provide analysis as to *when* this method helps. This could include an experiment that validates the claims about lower distribution shift error or an ablation on the properties of the datasets on which the approach works well.

Overall, I think this paper offers a neat new idea that can provide insights into how to build purely offline RL algorithms. However, I believe the theoretical section is the weakest part of the submission, and the paper might from this section being shortened. Further, precise notation is required should the authors intend to keep this section. The experiment section could be strengthened by additional analysis that helps understand when this method is useful. I do not think the claim that this method outperforms sota-algorithms is sufficiently supported. I do think that the paper provides an interesting new idea but in the current state I am recommending rejection.

[1] Michael J Kearns and Satinder P Singh. Finite-sample convergence rates for q-learning and indirect algorithms. NeurIPS 1999.

**Questions:**

Q1: Can you elaborate on assumption 4.3 and why this is a reasonable assumption to make?
Q2: Can you elaborate on what parts of section 4.3 are claimed to be novel in this work and which parts are taken from previous work?
Q3: Can you elaborate on what part of the theory is specific to your algorithm and which parts you believe are generally true for all approximate models?
Q4: Can you elaborate on why you chose standard deviation as a measure of dispersion?

---

> ### Author Response · Authors · 2024-11-24
> **Author Response 1**
>
> We thank you for your valuable feedback. We address your comments in the following.
>
> **Q1/weakness 1-d**: Assumption 4.3 is based on the theory that the diffusion world model prediction error, or intuitively the uncertainty (LHS of equation 8, 9) could be estimated by the diffusion ensemble variance, defined as the discrepancy (RHS of equation 8, 9). Similar assumptions are already used in classic model-based RL algorithms, such as the “assumption 4.3” in MOPO[1] and the “equation (3)” in MOReL[2], while both papers use MLP as the world model and bootstrap ensembles as the uncertainty estimator, and provide detailed theoretical and empirical analysis. Furthermore, the usage of bootstrap ensembles for uncertainty estimators has been justified theoretically in [3] and empirically in [4], while diffusion models also show a high correlation between uncertainty and discrepancy, presented empirically in recent papers [5-6]. Based on these facts, we believe that assumption 4.3 is a reasonable assumption. Besides, we didn’t give upper bounds for \alpha_m and \alpha_r in the paper, meaning that they could be set large enough to satisfy Equation 8 and Equation 9 for all possible states and actions.
>
> **Q2/weakness 1-a**: When using a world model to guide policy evaluation and learning, there are three sources of errors that may affect the resulting return gap, known as the model transition error, reward prediction error, and the policy distributional error. Our work is the first to consider all three sources of errors in diffusion world models and to quantify their joint impact on return. This result is stated in our main theorem. Existing work has only considered a subset of these errors. In particular, the work in [7] assumes that the reward function is known but in our paper, we take the reward prediction error and uncertainty penalty into account. Besides, MOPO[1] directly adds the uncertainty penalty to the reward, and MOReL[2] only uses it to determine the terminal signal. Our work provides a theoretical analysis with a new bound on the return gap.
>
> **Q3**: As we responded in Q2, the part of the uncertainty penalty on rewards is specific to our algorithm. If assumption 4.3 is removed, then the proof could lead to another looser bound which is generally true for all other models.
>
> **Q4**: Not sure we fully understand this question. The word “dispersion” is never used in our paper. If you mean “discrepancy”, it is defined in Section 4.1 as $d_\theta (s_t, a_t)$ and it’s not the standard deviation. Additionally, the same definition can be seen in MOReL, while MOPO did use the standard deviation as discrepancy.
>
> **Weakness 1-b**: We stated that $M$ is a Markov Decision Process (MDP) in the first sentence of Section 3, which includes transition probability $P$, initial state distribution $\mu _ 0$,  and other elements. Therefore, in Equation 5 $s_t$ is sampled from the marginal probability distribution of s in time step $t$ under MDP $M$ and policy $\pi$, with $s_t \sim P(\cdot |s_{t-1}, a_{t-1})$, $a_{t-1} \sim \pi( \cdot | s_{t-1})$ and $s_0 \sim \mu_0$. If it’s necessary, we can modify our paper and clarify this in the main text.
>
> **Weakness 1-c**: We don’t agree that equation 6 is ill-defined. The TV-distance is based on the joint distribution of s_t and a_t instead of the reward function itself.
>
> **Weakness 1-e**: Thanks for your advice. We will adjust the theoretical part to improve readability. Also in fact, we have stated that “we give an outline of our proof and display the details in the Appendix” in the first sentence of Section 4.3.
>
> **Weakness 2-a, b, c**: Thanks for the question. We modified our paper by bolding the statistically significant results instead of the max averages. The variances of baselines are not included at first because we follow the presentation style of existing works on diffusion-based offline RL [8-10] and try to make the table look more concise. We have added the variances of the baselines in Table 1 as the main results in the revision. The original papers of baselines in Table 2 didn’t provide their variances. Thus we left Table 2 unchanged and moved it to Appendix. It’s not surprising that the improvements achieved by our solution fluctuate across different environments and datasets. We would like to note that almost all previous work in this area has demonstrated similar (if not lower) improvements over existing methods. However, we need to point out that the average score is still significant in Table 1, which supports the claim that our algorithm outperforms other SOTA algorithms. Table 2 shows the performance of other environments with sparse reward functions, in which high variance is almost unavoidable.

---

> ### Author Response · Authors · 2024-11-24
> **Author Response 2**
>
> **Weakness 2-d:** We did analyze in the ablation study that the uncertainty penalty can improve the stability of training in all kinds of datasets, and importance sampling has more impact on suboptimal datasets such as “medium-replay” in which the distribution is more scattered while having limited effect on nearly optimal datasets. Besides, although we use the same hyper-parameters of uncertainty penalty for all datasets, the optimal values of these hyper-parameters vary among different datasets, since Figure 4 in the Appendix shows different scales of uncertainty and discrepancy in different datasets. However, an analysis of the dataset itself is not the emphasis of this paper.
>
> [1]Yu, Tianhe, et al. "Mopo: Model-based offline policy optimization." Advances in Neural Information Processing Systems 33 (2020): 14129-14142.
>
> [2]Kidambi, Rahul, et al. "Morel: Model-based offline reinforcement learning." Advances in neural information processing systems 33 (2020): 21810-21823.
>
> [3]Peter J Bickel and David A Freedman. Some asymptotic theory for the bootstrap. The annals of statistics, pages 1196–1217, 1981.
>
> [4]Kurtland Chua, Roberto Calandra, Rowan McAllister, and Sergey Levine. Deep reinforcement learning in a handful of trials using probabilistic dynamics models. In Advances in Neural Information Processing Systems, pages 4754–4765, 2018.
>
> [5]Berry, L., Brando, A. &amp; Meger, D.. (2024). Shedding Light on Large Generative Networks: Estimating Epistemic Uncertainty in Diffusion Models. Proceedings of the Fortieth Conference on Uncertainty in Artificial Intelligence, PMLR 244:360-376, 2024.
>
> [6] Shu, Dule, and Amir Barati Farimani. "Zero-Shot Uncertainty Quantification using Diffusion Probabilistic Models." arXiv preprint arXiv:2408.04718 (2024)
>
> [7] Janner, Michael, et al. "When to trust your model: Model-based policy optimization." Advances in neural information processing systems 32 (2019).
>
> [8] Janner, Michael, et al. "Planning with Diffusion for Flexible Behavior Synthesis." International Conference on Machine Learning. PMLR, 2022.
>
> [9] Ajay, Anurag, et al. "Is Conditional Generative Modeling all you need for Decision Making?." The 11th International Conference on Learning Representations (ICLR), 2023.
>
> [10] Wang, Zhendong, Jonathan J. Hunt, and Mingyuan Zhou. "Diffusion Policies as an Expressive Policy Class for Offline Reinforcement Learning." The 11th International Conference on Learning Representations (ICLR) 2023.

---

> ### Comment · Reviewer_TBhS · 2024-11-25
>
> Dear authors,
>
> thank you for the clarifications and the thorough response. I have also read the other reviews. It seems that reviewer PaFJ had similar concerns about the clarity of section 4 and questions the dependence on diffusion models. Reviewer g96b seems to have similar concerns about statistical validity.
>
> I appreciate you working on updating the statistical significance in the plots.
> Also, thank you for explaining the notation of the expectation. I read the commata as dividers. I believe it might be clearer if parenthesis are used around (s, a).
> After reading the responses to my questions, I will provide some more detail to my review.
>
> **Assumption 4.3.** The reason why I asked specifically about this assumption is because it bounds the TV distance of 2 distributions by the expected distances over *states*. This seems to disregard the presence of aleatoric uncertainty and only works with epistemic uncertainty. This is the reason why I am also asking about what benchmarks the method is applicable to. I believe the current manuscript does not differentiate between these uncertainties. The model’s performance could be perfectly accurate but the distance between two next states could still be arbitrarily large since there are no assumptions on the MDPs that are being considered. Thus, it seems that this method can only work in deterministic MDPs or possibly MDPs with Lipshitz transitions which would require proof. Lipshitz transitions which have been studied before [1]. This would be a big limitation and should be addressed in future versions of the paper.
>
> An example: Consider an MDP where the largest reward can be obtained in using a state-action pair whose true distribution over next states has high variance. The method would actively penalize the reward for going to these high reward states.
>
> Other works attempt to get around this issue by obtaining a distribution $Q$ over transitions $P$ and measuring the variance of that distribution $Q$. That is the ensemble methods that are being referred to in the manuscript. I don’t think it is easily possible to achieve such estimates by sampling from the same distribution $P$.
>
> **Theoretical Bounds** First, as I mentioned, the theory of sample and computation complexity is old and goes back at least 25 years. I do not agree that a method from 2020 is a classical method. After a closer look I realized that this paper only cites work from 2018 and newer. If the manuscript ought to make a strong theoretical contribution, I recommend it include relevant literature that covers model-based sample complexities and error bounds which would allow for comparisons. Lemmata such as 4.7 and 4.8 are standard tools in these works and the presented manuscript does not make it clear that these tools exist and are being reused.
>
> On the novelty of the result: It is well known that if the TV distance on transitions and returns can be bounded, one can achieve low error. Again this literature goes back to at least the paper I cited from 1999. A fundamental challenge is to develop algorithms that effectively minimize this distance.
>
> 1) The presented work assumes that the error is bounded by some arbitrarily large quantity. Given a high dimensional state-space, the distance between two states generated by a random model ought to be larger than 1 which would make the assumed bound vacuus. I can always get a guarantee that the difference in returns is lower bounded by the maximum return. (i.e. if the TV distance is bounded by 1 this is basically the second term on the RHS of equation 11). Stating that alpha can be picked arbitrarily large does not resolve this issue; it might even make it worse.
>
> 2) In order to make a theoretical contribution, it would be relevant to demonstrate that the presented method can minimize the true TV distance which I am not convinced of. This is due to the issues with uncertainty that I elaborated on earlier.
>
> 3) The reward is often assumed to be known as estimating the reward is information theoretically easy if one can estimate the transitions. Usually, no novel technical extensions are needed to include it.
>
> **Equation 6** I’m not sure what joint distribution the rebuttal is referring to. In the manuscript R is defined as a function. If the notation is being overloaded here, that should probably be clarified.
>
> **Ablation study** I appreciate the pointer to the ablation study but it is still a study on maximum rewards. To give an example for the point that I raised. Since the paper is trying to mitigate o.o.d. estimation, it would be interesting to see how for instance Q-values of o.o.d. state-action pairs behave. This would be an experiment related to reviewer g96b's point on overestimation.
>
> **Dispersion** I was referring the the variance measure in the experiments. The paper reports mean and standard deviation. I was asking why standard deviation is used.
>
> [1] Lipschitz Continuity in Model-based Reinforcement Learning. Kavosh Asadi et al.,ICML 2018

---

> > ### Author Response · Authors · 2024-11-27
> >
> > > if parenthesis are used around (s, a)
> >
> > Thanks for this suggestion. We have implemented it in our revision. Please refer to our new version uploaded.
> >
> > >Assumption 4.3
> >
> > Thanks for the comment on aleatory uncertainty and epistemic uncertainty. As you mentioned, there is existing work that addresses this issue by obtaining a distribution Q over transitions P and measuring the variance of that distribution Q. There has been work using a similar idea with a single diffusion model and a Bayesian hyper-network. The proposed solution in [1] generates an ensemble of predictions using this method and achieves the desired estimates. It can be easily added to our proposed solution. Such uncertainty estimation using standard deviation and an appropriately tuning scale parameter is commonly and successfully applied in many works of offline RL[2-3] though lacking theoretical analysis. Given the time left and the fact that our solution is already achieving solid improvements over the baselines, we plan to clarify it in our paper (e.g., MDPs with Lipschitz transitions) and mention this as future work. Please also refer to our response below regarding the theoretical analysis. Based on the reviewer’s comments, we believe our main theorem can be updated to provide a better explanation/justification of our approach. It will also help address the issue here.
> >
> > >Theoretical Bounds
> >
> > Thanks for the further comments. In fact, we could drive our main theorem without using the maximum TV distances defined in Definitions 4.1 and 4.2. Theorem 4.5 will remain the same by replacing $\hat {\varepsilon} _ r(\pi)$ by an expectation $L_1= \mathbb{E} _ {(s_t , a_t) \sim \pi \vert \mathcal{M}} \left[ \vert R(s_t,a_t) - r_\eta(s_t,a_t) \vert \right]$ and replacing  $\hat{\varepsilon} _ m(\pi)$  by $ L_2=\mathbb{E} _ {(s_t, a_t) \sim \pi \vert \mathcal{M}} \left[D_{TV} (P(s_{t+1}\vert s_t,a_t) \Vert P_\theta(s_{t+1} \vert s_t,a_t))\right]$, where the expectations are both taken over samples drawn from the target policy $\pi$.
> >
> > As the reviewer pointed out,  a fundamental challenge is to develop algorithms that effectively minimize these distances. The updated theorem using $L_1$ and $L_2$ is exactly what inspired our design of IS. More precisely, let $\pi_b$ represent the behavior policy used to collect offline data. It is easy to show that $L_1$ and $L_2$ in the updated theorem can be written as $L_1 = \mathbb{E} _ {(s_t, a_t) \sim \pi _ \mathcal{D}\vert \mathcal{M}} \left[\frac{\pi(s_t, a_t)}{\pi _ {\mathcal{D}} (s_t,a_t)} \left[\vert R(s_t, a_t) - r_\eta(s_t,a_t)\vert \right]\right]$ and $ L _ 2 = \mathbb{E} _ {(s_t, a_t) \sim \pi _ {\mathcal{D}} \vert \mathcal{M}}  \frac{\pi(s_t,a_t)}{\pi_ {\mathcal{D}} (s_t,a_t)} \left[D_{TV}(P(s_{t+1}\vert s_t,a_t) \Vert P_\theta(s_{t+1} \vert s_t,a_t))\right]$, where both expectations are taken over samples drawn from the behavior policy $\pi _ \mathcal{D}$ instead, but with IS re-weighting $\frac{\pi(s_t,a_t)}{\pi_ {\mathcal{D}} (s_t,a_t)}$. Our proposed algorithm is exactly minimizing such IS re-weighted $L_1$ and $L_2$, thus minimizing the return gap characterized in the updated Theorem 4.5.
> >
> > This provides a much more clearer explanation of our results. It now demonstrates that the presented method can indeed minimize the true TV distance terms in return gap. On the other hand, existing methods complete diffusion model training (using offline data) prior to policy evaluation. Taking the $L_1$ term as an example, existing methods are instead minimizing a different expectation $ G _ 1= \mathbb{E} _ {(s_t, a_t) \sim \pi _ \mathcal{D}\vert \mathcal{M}} \left[D_{TV}(R(s_t,a_t) \Vert r_\eta(s_t,a_t))\right]$ over samples drawn from $\pi _ \mathcal{D}$. We can see that the true loss function can be viewed as $L_1 = G_1 + <\pi-\pi _ \mathcal{D}, D_{TV}(R(s_t,a_t) \Vert r_\eta(s_t,a_t))> $. Thus existing methods minimizing $G_1$ is only considering a partial objective of the true loss function $L_1$.
> >
> > We sincerely thank the reviewer for the insightful discussion! In the previous version, we tried to state our results in a more succinct manner using the maximum TV distances $\hat{\varepsilon}_r(\pi)$ and $\hat{\varepsilon}_m(\pi)$. But it turned out to make it much harder to demonstration our theoretical contribution that inspired the proposed algorithm. Thank you again for the discussions!

---

> > ### Author Response · Authors · 2024-11-27
> >
> > > Equation 6
> >
> > To avoid confusion, we have changed that equation and all the related parts in the revision.
> >
> > > Ablation Study
> >
> > We did an extra ablation study on the OOD Q-values and attached it to the Appendix of our work.
> >
> > > Dispersion
> >
> > That’s because all the baselines report their results with standard deviations. It would be convenient and clear for comparison to follow this convention. However, we could consider other statistics recommended by the reviewer.
> >
> > [1] Chan, Matthew Albert, Maria J. Molina, and Christopher Metzler. "Estimating Epistemic and Aleatoric Uncertainty with a Single Model." The Thirty-eighth Annual Conference on Neural Information Processing Systems.
> >
> > [2] Sun, Yihao, et al. "Model-Bellman inconsistency for model-based offline reinforcement learning." International Conference on Machine Learning. PMLR, 2023.
> >
> > [3] Bai, Chenjia, et al. "Pessimistic Bootstrapping for Uncertainty-Driven Offline Reinforcement Learning." International Conference on Learning Representations. 2021.

---

> > > ### Comment · Reviewer_TBhS · 2024-11-27
> > >
> > > Dear authors, thank you for your thorough response and the hard work you have put in. I believe that this method may have technical empirical merit for domains such as robotics and the results are encouraging. However, I do not believe that the manuscript in its current stage is ready for publication. **I will summarize this opinion here for the AC.**
> > >
> > > * The paper presents a general purpose offline RL method but the method is limited as it can only be effectively applied to deterministic MDPs. This needs to be clear from the beginning and I believe the paper should not present the method as a general RL method. It might work in MPDs with Lipshitz transitions but that would require proof.
> > > * The theoretical results are not convincing and I believe they take away more from the paper than they add by confusing the reader. It is not clear which parts are novel and I believe it is well known that one can bound the return if one can bound the TV distance of the transitions and reward.
> > > * The new manuscript still contains some (potentially) incorrect statements/sections that that have even been added or not been fixed.
> > >   * L232: "since each output is denoised from Gaussian noise, the uncertainty could be directly estimated from the
> > > discrepancy of multiple denoised samples with the same condition" - This statement is false in an MDP that has stochastic transitions. As a result, Assumption 4.4 would not be satisfied in a stochastic MDP.
> > >   * L394: "These two loss function are taking over samples drawn from the behavior policy, but with importance sampling reweighting π(st,at)/πD (st,at) , thus minimizing the return gap C directly." - These two objectives do not directly minimize the return gap. They minimize weighted versions of the parameters in the return gap. Whether or why this actually minimizes $\hat{\varepsilon}_p$ is not clear to me from the text.  $\hat{\varepsilon}_p$ would for instance be minimized by behavior cloning and that is exactly not the goal in RL. It could also be that the importance sampled optimization of $P$ and $R$ leads to worse bounds on the respective errors of these objects. Consider for instance a policy that only visits some state very infrequently. I do not see why the error on that transition should be small if that state is downsampled a lot, especially when things are stochastic. To obtain guarantees for the latter, it might make sense to look at ideas related to [1].
> > >   * Theorem $4.5$ does not have any algorithm dependent parameters and it is not clear to me that it is novel. It seems that the importance sampling play little to no role in the proofs.
> > > * The paper lacks a fundamental discussion of existing theoretical bounds from the literature if it aims to make a theoretical contribution.
> > >
> > > As a result, I will maintain my score and recommend rejection.
> > >
> > > [1] Minimax-Regret Sample Selection in Randomized Experiments. Yuchen Hu, Henry Zhu, Emma Brunskill, Stefan Wager. 2024.

---

### Official Review · Reviewer_PaFJ · 2024-11-05

**Soundness:** 3
**Presentation:** 3
**Contribution:** 3
**Rating:** 6
**Confidence:** 3

**Summary:**

This work attempts to address the important issue of policy distribution shift in offline RL. The authors propose a novel method which uses a world model as a synthetic data generator for closed-loop policy evaluation, where the world model is adapted via importance sampling to the changing distribution of the learned policy. The proposed method is supported be theoretical bounds on the return gap and shows impressive performance on key offline tasks with suboptimal offline datasets.

\* Note that while this paper does fall squarely within my expertise, I was not able to give it the time it deserves and that is reflected in my confidence score.

**Strengths:**

1. The proposed method is novel, well-motivated and clearly explained.
2. Theoretical results to support claims of bounding the return gap between world model and environment.
3. Strong results.

**Weaknesses:**

1. While I like the method proposed by the paper, I don't see a clear dependency on diffusion. From my understanding, the method can be generalized to any world model with some form of uncertainty estimation. As such, I think the paper would be stronger if the method is generalized to any world model. However, I would also be satisfied by an ablation comparing diffusion to other world model types, and clearly showing why diffusion is necessary for the level of performance presented.
2. The paper can be made shorter and more concise to improve readability. I believe Section 3 is mostly unnecessary. You introduce the full notation and background of diffusion but barely sue it in the main text. I recommend shortening it significantly (possibly including it in Section 4) and leaving the full notation and explanation in the appendix as it is necessary for the proofs.
3. In Section 4.1, you state that "Introducing $s_{t+1}$ as an extra input significantly improves accuracy of reward prediction [...]". This is atypical in world model literature and I would recommend backing up this claim with an ablation. Terminating based on high uncertainty is also new to me and a great suggestion! I would also love to see an ablation of this as this changes the distribution of your trajectories significantly.
4. The main results in Table 1 are poorly presented. I would recommend replacing the table with aggregated statistics as suggested by [(Agarwal et al, 2022)](https://arxiv.org/abs/2108.13264). Table 2 can also be bundled into this figure.


Minor remarks:
1. I found Figure 1 insufficient to grasp the proposed method. I was only able to grasp it after I read the full work at which point the figure has little value. I recommend using a simpler graph with only a few data points and simpler text annotations within the figure itself. (b3) is unnecessary, only showing the (b1) -> (b2) will improve readability.
2. Figure 2 can also be mode more self-explanatory and independent. The two replay buffers are confusing in the figure as they aren't sufficiently explained. The diffusion steps do not necessarily need to be visualized. You probably don't need to spell out each variable being sampled from the buffers. I would also suggest changing the blue box to 'Policy Evaluation within World Model'.
3. Line 228, (I assume) missing 'Section 3'.
4. Line 279, unclear what the loss $l$ is at this point of reading the text.
5. Figure 3, keep y axis the same between all subfigures. Move legend under figures. Possibly remove 'random' as it does not add to the paper.

**Questions:**

1. I like the idea of using IS to adapt the world model but the reasoning is not exactly clear to me. Usually, we want to use IS to estimate some variable under an unknown distribution by reweighing it by some other known distribution. However, in this case, we can estimate both distributions very well. If my understanding is correct, can you motivate the use of IS more?
2. From my understanding, IS is a poor technique to use when the two (policy) distributions are very different and that is a completely plausible scenario in your problem setting. Can you explain how you avoid this? Furthermore, have you considered alternative techniques such as MCMC?

---

> ### Author Response · Authors · 2024-11-24
> **Author Response 1**
>
> We thank you for your insightful and encouraging review, which inspires us for our future research. We address your concerns in the following.
>
> **Weakness 1**: The adoption of diffusion models in this paper is mainly to utilize its denoising procedure to estimate the uncertainty and augment the generalization ability. The uncertainty estimation implies is highly correlated with the prediction error of the diffusion model. By applying penalties on uncertainty, the target policy is prevented from exploring unseen states and action pairs with overestimated returns. The method of uncertainty estimation in this paper is exclusive to diffusion models. Nevertheless, it’s true that other world model types with traditional uncertainty estimation methods such as ensembles and dropouts could also be combined with importance sampling. We would consider it as a future work and add some discussions in the paper. Thanks for your valuable suggestion.
>
> **Weakness 2**: Thanks for your suggestion, we have shortened Section 3 in the revision to provide more readability.
>
> **Weakness 3**: We are glad that you noticed these. The effectiveness of these two mechanisms is well supported by our experimental results. We test the average reward prediction error over 100k samples, random policy, and the same diffusion world model with a horizon of 5, and show the results as follows:
> | envs | $r(s_t, a_t, \hat{s}_{t+1})$ | $r(s_t, a_t)$ |
> |-----------|-----------|-----------|
> |halfcheetah-medium| $0.067\pm0.012$ | $0.098\pm0.011$ |
> |halfcheetah-medium-replay| $0.093\pm0.015$ | $0.135\pm0.011$ |
> |halfcheetah-medium-expert| $0.098\pm0.009$ | $0.126\pm0.018$ |
> |hopper-medium| $0.008\pm0.001$ | $0.013\pm0.002$ |
> |hopper-medium-replay| $0.008\pm0.001$ | $0.009\pm0.001$ |
> |hopper-medium-expert| $0.006\pm0.001$ | $0.010\pm0.002$ |
> |walker2d-medium| $0.071\pm0.005$ | $0.084\pm0.004$ |
> |walker2d-medium-replay| $0.060\pm0.008$ | $0.076\pm0.006$ |
> |walker2d-medium-expert| $0.064\pm0.004$ | $0.076\pm0.008$ |
>
> Based on these results, we found that introducing $\hat{ s } _ {t+1}$ into the reward model could largely reduce the prediction error in all datasets by 10% to 40%. A possible explanation for this is that in most environments, including Mujoco locomotion tasks, the true reward functions are defined directly with $s_t$ and $s_{t+1}$. One common example is defining the moving distance between two states as the reward. Therefore, by introducing $s_{t+1}$ in the input, the trained reward model becomes more similar to the true reward function, thus has more generalizing capability.
>
> The terminating based on high uncertainty is first introduced in MOReL[1] for offline reinforcement learning, and it’s also a crucial component to ensure training stability in our algorithm. Figure 3 in the ablation study shows that without uncertainty penalty (including termination), the training fails in all medium, medium-replay, and medium-expert datasets in halfcheetah environment. Same phenomenon is observed in hopper and walker2d environments. We also notice that a threshold too low also leads to poor performance, since the synthetic trajectories are strictly constrained near the distribution of offline dataset.
>
> **Weakness 4**: Thank you for this suggestion. We add the aggregated statistics for both Table 1 and the ablation study in the revision.  However, since D4RL is a widely used benchmark in offline reinforcement learning, it would be convenient for the community to compare the performance with exact numbers present in the form of tables. Therefore, we decided to keep Table 1 and Table 2 like many previous works did and move Table 2 to Appendix.
>
> **Minor Remarks**: We appreciate the reviewer for highlighting these, we adjusted Figure 1 and Figure 2 to make it easier to understand. Figure 3 is changed into aggregated statistics including the results in the other two environments. The mentioned typos are also addressed in the revision.

---

> ### Author Response · Authors · 2024-11-24
> **Author Response 2**
>
> **Questions**:
> >I like the idea of using IS to adapt the world model but the reasoning is not exactly clear to me. Usually, we want to use IS to estimate some variable under an unknown distribution by reweighing it by some other known distribution. However, in this case, we can estimate both distributions very well. If my understanding is correct, can you motivate the use of IS more?
>
> This is a good point. Our method is indeed estimating a form of expectation under an unknown distribution. We can consider two trajectory distributions $P_b$ and $P_t$ in the real environment: $P_b$ is obtained by the behavior policy to collect the offline dataset, and $P_t$ is obtained by the target policy we are optimizing. When training the diffusion model, we should minimize an expected loss by sampling trajectories according to $P_t$, so that the learned diffusion model can minimize the error of evaluating the target policy. However, since P_t is not known during data collection and training, the diffusion model is trained using data sampled according to $P_b$ instead. As the target policy further deviates from the behavior policy during the training, this problem becomes more serious. As pointed out by the reviewer, our algorithm leverages IS to address this problem and reweight the samples based on $P_t$. It enables continual alignment between the target policy and world model during training.
>
> >From my understanding, IS is a poor technique to use when the two (policy) distributions are very different and that is a completely plausible scenario in your problem setting. Can you explain how you avoid this? Furthermore, have you considered alternative techniques such as MCMC?
>
> Main reasons: 1. IS has been adopted by many existing policy-based offline RL algorithms. It is a simple yet effective solution. We are using IS on the world model to support continual alignment with the target policy. 2. The use of the uncertainty penalty could constrain the policy from deviating too much away from the behavior policy since unseen state and action pairs will get punishment.
>
> Tricks: We set a maximum for the IS weight.
>
> An alternative technique using MCMC seems a very interesting idea, we will consider it in our future research. Thanks for this suggestion.
>
> [1] Kidambi, Rahul, et al. "MOReL: Model-based offline reinforcement learning." Advances in neural information processing systems 33 (2020): 21810-21823.

---

> ### Comment · Reviewer_PaFJ · 2024-12-02
>
> > Weakness 2: Thanks for your suggestion, we have shortened Section 3 in the revision to provide more readability.
>
> I do not see substantial changes in the revised PDF.
>
> > Weakness 3: We are glad that you noticed these. The effectiveness of these two mechanisms is well supported by our experimental results. We test the average reward prediction error over 100k samples [...]
>
> Thank you for sharing the results and your following hypothesis. Your statement makes the subtle assumption that more accurate reward prediction will result in better policy learning. While a fair assumption, it is still unclear if it is a valid one [1] [2]. As such, I am not convinced and would like to see an ablation.
>
> > Weakness 4: [..] However, since D4RL is a widely used benchmark in offline reinforcement learning, it would be convenient for the community to compare the performance with exact numbers present in the form of tables.
>
> In my opinion, poor historical choices shouldn't be a valid reason for poor future decisions. The new figures used in the revised version (e.g. Fig 3) significantly increase the readability of your work, and I urge you to do the same for Table 1. You can still keep the table in the appendix to make it more easily comparable with prior work. This comment does not affect my score.
>
> > This is a good point. Our method is indeed estimating a form of expectation under an unknown distribution. We can consider two trajectory distributions $P_b$ and $P_t$ in the real environment [..]
>
> This point is still unclear to me. I understand that you can treat $P_t$ as an unknown distribution and estimate it via IS. However, what is stopping you from estimating $P_t$ by simply rolling out the policy within your world model?
>
> > Main reasons: 1. IS has been adopted by many existing policy-based offline RL algorithms. It is a simple yet effective solution. We are using IS on the world model to support continual alignment with the target policy. 2. The use of the uncertainty penalty could constrain the policy from deviating too much away from the behavior policy since unseen state and action pairs will get punishment.
>
> Thank you for the answer. Unfortunately, I do not find (1) to be valid reasoning, and (2) appears to be an unverified hypothesis.
>
> I will keep my score as is for the time being.
>
> [1] Lambert et al. (2020) Objective Mismatch in Model-based Reinforcement Learning
>
> [2] Wei et al. (2023) A Unified View on Solving Objective Mismatch in Model-Based Reinforcement Learning

---

### Author Response · Authors · 2024-11-24
**Official Comment by Authors**

Dear all reviewers,

We would like to thank the reviewers for taking their time and providing their insightful feedback. We're still working on the revision and will bring it in the next one or two days. Once we get it done, we'll make a new comment and also provide a summary of modifications.

---

> ### Author Response · Authors · 2024-11-27
> **Revision uploaded**
>
> Dear all reviewers,
>
> We have updated the revision of our paper. For ease of comparison, we highlight all major changes in blue.
>
> **Summary of revisions:**
>
> Shorten Section 3.
>
> Rewritten the theoretical part and the proof in the Appendix.
>
> Adjust Figure 1 and Figure 2 as suggested by reviewer PaFJ and g96b.
>
> Add the variances of baselines in Table 1.
>
> Use aggregated statistics to report the results, and extend the compared environments of ablation study to hopper and walker2d.
>
> Add the results of all required additional experiments to the appendix.
>
> Fix all the minors mentioned by the reviewers.
>
> Thanks again to all of the reviewers for providing valuable feedback and helping us improve our work.

---

### Meta-Review · Area_Chair_LUpw · 2024-12-14

**Metareview:**

This paper presents a model-based offline method that uses an uncertainty-penalized diffusion model to evaluate the evolving policy and keep the policy within the support of the offline data. The authors present both theoretical results that upper bound the return gap between their method and the real environment under the target policy and empirical results on D4RL.

Reviewers found the writing clear, the related work discussion mostly complete, and empirical results and ablations thorough. However, there were concerns that the theoretical results are unconvincing and not properly situated within the existing literature. There are also several problematic assumptions and statements in the paper even after the revision post-rebuttal, and therefore the paper is not ready for acceptance. We urge the authors to take this constructive feedback into consideration and re-submit a revised version to a future venue.

**Additional Comments On Reviewer Discussion:**

During the rebuttal and reviewer discussion phase, although the authors did satisfy some concerns, the reviewers

Reviewer PaFJ was one of the higher scores (at 6) but did not feel the authors had presented a strong enough case during the rebuttal, mostly due to weak arguments and poor assumptions. Taking into account the final feedback from reviewer TBhS, they decided not to fight for acceptance.

Reviewer TBhS raised several concerns, including that the theoretical contributions are seen as unconvincing and lacking novelty, with some familiar results presented in a confusing manner. Additionally, there are potentially incorrect or unclear statements, such as those regarding denoising Gaussian noise in stochastic MDPs and the loss functions not directly minimizing the return gap. The theorem also lacks algorithm-dependent parameters and the role of importance sampling in the proofs is unclear. Finally, the manuscript lacks a discussion of existing theoretical bounds from the literature.

---

### Decision · Program_Chairs · 2025-01-22

Reject